# Enhancing Graph Contrastive Learning for Protein Graphs from Perspective of Invariance

Yusong Wang [* 1 2]  Shiyin Tan [* 2]  Jialun Shen [* 2]  Yicheng Xu [2]  Haobo Song [3]  Qi Xu [4]  Prayag Tiwari [5]
Mingkun Xu [† 1]

## Abstract

Graph Contrastive Learning (GCL) improves Graph Neural Network (GNN)-based protein representation learning by enhancing its generalization and robustness. Existing GCL approaches for protein representation learning rely on 2D topology, where graph augmentation is solely based on topological features, ignoring the intrinsic biological properties of proteins. Besides, 3D structure-based protein graph augmentation remains unexplored, despite proteins inherently exhibiting 3D structures. To bridge this gap, we propose novel biology-aware graph augmentation strategies from the perspective of invariance and integrate them into the protein GCL framework. Specifically, we introduce Functional Community Invariance (FCI)-based graph augmentation, which employs spectral constraints to preserve topology-driven community structures while incorporating residue-level chemical similarity as edge weights to guide edge sampling and maintain functional communities. Furthermore, we propose 3D Protein Structure Invariance (3-PSI)-based graph augmentation, leveraging dihedral angle perturbations and secondary structure rotations to retain critical 3D structural information of proteins while diversifying graph views. Extensive experiments on four different protein-related tasks demonstrate the superiority of our proposed GCL protein representation learning framework.

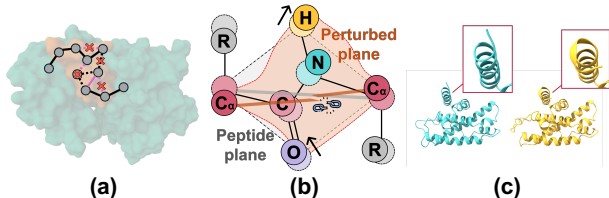

*Figure 1.* Visualization of limitations in existing graph augmentation approaches for protein representation learning. Figure (a) shows a functional domain where nodes and edges are erroneously removed during graph augmentation. Figure (b) depicts how traditional 3D augmentation with coordinate pernturbation distorts the peptide plane and critical bond relationships. Figure (c) compares the original protein structure (left) and its augmented views (right) generated by SWISS-MODEL, a protein homology modeling tool. High similarity limits diversity.

## 1. Introduction

Graph Neural Networks (GNNs) (Scarselli et al., 2009) enable efficient protein representation learning by modeling proteins as graphs, where residues represent nodes and bonds represent edges. GNN encoders map each protein into a vector representation, which is then processed by a prediction module to predict specific properties (Zhang et al., 2023; Wang et al., 2023b). Graph Contrastive Learning (GCL) enhances GNN-based protein representation learning in two ways: first, it improves the generalization and robustness of GNNs through diverse protein graph augmentation strategies; second, GCL effectively captures structural patterns across different protein conformations, leading to more discriminative protein representations (Li et al., 2022; Wang et al., 2022a; Tan et al., 2024).

Current GCL approaches for protein representation learning focus on 2D topology-based graph augmentation (Suresh et al., 2021). They seek to minimize redundant information while preserving essential information across augmented views, assuming critical protein structures remain unchanged (Suresh et al., 2021). For instance, Wei et al. (2023) identify important edges between residues in a graph by measuring their semantic impacts based on the gradients of mutual information. Liu et al. (2022a) explores graph

---

[*]Equal contribution [1]Guangdong Institute of Intelligence Science and Technology, Zhuhai, China [2]School of Engineering, Institute of Science Tokyo, Tokyo, Japan [3]Graduate School of Medicine, The University of Tokyo, Tokyo, Japan [4]School of Computer Science and Technology, Dalian University of Technology, Dalian, China [5]School of Information Technology, Halmstad University, Halmstad, Sweden. Correspondence to: Mingkun Xu† <xumingkun@gdiist.cn>.

*Proceedings of the 42nd International Conference on Machine Learning*, Vancouver, Canada. PMLR 267, 2025. Copyright 2025 by the author(s).

augmentation in the spectral space to preserve 2D protein topology through eigen-value-based transformations. However, only focusing on topological structures tends to **neglect the inherent biological significance of proteins**, which dilutes or even discards structural-functional associations tied to protein functionality. As Figure 1 (a) shows, such graph augmentation disrupts a functional domain which is essential for protein function (Heinemann et al., 2021).

Furthermore, while integrating 3D structural information can enhance protein representation learning (Liu et al., 2022b), **3D structure-based graph augmentation remains unexplored**. Although several 3D augmentation strategies exist, they face challenges when applied to GCL for protein representation learning: (1) applying traditional 3D augmentation (*e.g.*, random coordinate perturbations, anisotropic scaling) (Hermosilla et al., 2021; Wang et al., 2023b) risks protein structure disruption. As Figure 1 (b) shows, the peptide plane is distorted by random coordinate perturbations, affecting protein representation learning. (2) using protein homology modeling tools like MODELLER (Eswar et al., 2006) and SWISS-MODEL (Waterhouse et al., 2018) to generate conformations (Li et al., 2022; Gao et al., 2023) reduces structural diversity. As Figure 1 (c) shows, generated structures are nearly identical to original structures. It could also introduce inaccurate protein structure (see Appendix A). Applying GCL with these compromised augmentation graphs can result in learning representations of biologically irrelevant proteins, undermining the model's performance on downstream tasks.

To address aforementioned issues, we incorporate biological properties of proteins to guide graph augmentations, thereby helping the GCL model learn accurate protein representations. Specifically, we propose two biology-aware graph augmentation strategies from the perspective of (i) 2D topology-based Functional Community Invariance (FCI) and (ii) 3D Protein Structure Invariance (3-PSI) and integrate them into a unified GCL framework for protein representation learning. FCI ensures that functional communities, *i.e.*, clusters of spatially and chemically similar residues collaborating to enable specific functions, remain intact. It considers both topological structures and chemical properties, ensuring the preservation of proteins' functional communities during graph augmentation. To generate augmented graphs while preserving 3D-related information of protein, 3-PSI uses two distinct coordinate perturbation strategies: (1) rotations of backbone dihedral angles, and (2) rotations of secondary structures ($\alpha$-helices and $\beta$-sheets), preventing the disruption of peptide planes and secondary structures during graph augmentation.

Main contributions of this research are summarized as:

- Based on the perspective of invariance, we propose two novel biology-aware graph augmentation strategies to

preserve biological integrity within proteins.

- We develop a unified protein GCL framework that integrates both graph augmentation strategies to facilitate protein representation learning.

- Experimental results on four downstream protein-related tasks highlight the superior performance of our proposed GCL framework, demonstrating its effectiveness in learning protein representations.

## 2. Related Work

**Protein representation learning.** Protein representation learning methods can be broadly divided into two categories: sequence-based and structure-based approaches (Quan et al., 2024). Sequence-based methods primarily utilize word embedding techniques to capture the semantic relationships between residues (Asgari & Mofrad, 2015; Yang et al., 2018; Zhuo et al., 2024) and one-dimensional convolutional neural networks to extract local sequence patterns and motifs (Hou et al., 2017; Tsubaki et al., 2018; Kulmanov & Hoehndorf, 2019). For example, ProtBERT-BFD (Elnaggar et al., 2020a) trained a BERT model on a large corpus of protein sequences and demonstrated superior performance by capturing sequential dependencies and evolutionary relationships between residues. Structure-based approaches, predominantly based on GNN, are more effectively leverage structural information, such as spatial relationships, than sequence-based methods, enabling a more comprehensive understanding of protein structures (Zhang et al., 2023; Wang et al., 2023b; Jamasb et al., 2024). For instance, GraphTrans (Liu et al., 2022b) developed a graph transformer architecture that processes protein structures as spatial graphs to achieve state-of-the-art (SOTA) performance in protein property prediction tasks. Our research focuses on the latter.

**Graph Contrastive Learning.** GCL has been widely adopted for enhancing GNN-based protein representation learning (Li et al., 2022; Tan et al., 2024; Wang et al., 2025). Current approaches focus on 2D topology-based graph augmentations, altering graph structures through node and edge modifications. Early methods, such as random node/edge dropping (Zhu et al., 2020; You et al., 2020; 2021) often disrupted semantic integrity of proteins by removing key residues and bonds. Recent methods enhance protein semantic preservation via learnable distributions, mutual-information gradients, adversarial training, and spectral constraints (Yin et al., 2021; Wei et al., 2023; Suresh et al., 2021; Liu et al., 2022a; Lin et al., 2023a). However, their focus on topology overlooks the biological significance of functional communities, missing key protein features.

Despite proteins' inherent 3D nature, 3D structure-based protein graph augmentations remain unexplored. Existing

3D augmentation strategies face challenges when applied to GCL for protein representation learning: traditional 3D augmentation methods, such as coordinate perturbations and anisotropic scaling (Hermosilla et al., 2021; Wang et al., 2023b; Feng et al., 2024), often disrupt peptide planes and protein hierarchical structures, yielding biologically implausible proteins; protein homology modeling tools (*e.g.*, MODELLER (Eswar et al., 2006) and SWISS-MODEL (Waterhouse et al., 2018)) generate structural variants as augmented views (Li et al., 2022) with limited diversity or inaccuracies, undermining the model's performance.

In this work, we incorporate inherently invariant properties of proteins into the augmented graphs and propose two graph augmentation strategies: FCI and 3-PSI, improving protein representation learning.

## 3. Preliminary

We employ GNN for its flexibility in integrating protein topology and geometry.

**Notations.** Let $G = (\mathcal{V}, \mathcal{E}, \mathcal{P})$ be a protein graph with $n$ nodes (residues, each represented by its central carbon atom $C_\alpha$) and $m$ edges, where $\mathcal{V} = \{v_i\}_{i=1}^n$ denotes the set of nodes, and $\mathcal{E} = \{e_{ij}\}$ represents the set of edges. $\mathcal{P} = \{P_i\}_{i=1,\dots,n}$ denotes the set of position matrices, where each $P_i \in \mathbb{R}^{k_i \times 3}$ represents the position matrix for node $i$. The protein graph construction, including node features, edge features and rotational invariance, are detailed in Appendix C.1. The graph can be represented as adjacency matrix $\mathbf{A} \in \{0, 1\}^{n \times n}$, where $\mathbf{A}_{i,j} = 1$ if an edge exists between node $i$ and $j$, otherwise $\mathbf{A}_{i,j} = 0$. The normalized Laplacian matrix is defined as $\mathbf{L}_{\text{norm}} = \text{Lap}(\mathbf{A}) = \mathbf{I}_n - \mathbf{D}^{-1/2}\mathbf{A}\mathbf{D}^{-1/2}$, where $\mathbf{I}_n$ is an identity matrix, $\mathbf{D}$ is a degree matrix, $\mathbf{D}_{i,i}$ represents the degree of node $i$, and $\mathbf{D}_{i,j} = 0$ if $i \neq j$. The complement of adjacency matrix is represented as $\mathbf{A}^c$. $\mathbf{A}^c_{i,j} = 1$ if an edge does not exist between node $i$ and $j$, otherwise $\mathbf{A}^c_{i,j} = 0$.

**Problem definition.** A protein is represented as a quadruplet $G = (\mathcal{V}, \mathcal{E}, \mathcal{P}, \mathcal{Y})$. $\mathcal{Y}$ is the set of labels. The objective of protein classification is to learn a mapping $f : (\mathcal{V}, \mathcal{E}, \mathcal{P}) \rightarrow \mathcal{Y}$.

**Graph Contrastive Learning.** GCL trains an encoder to maximize the mutual information between an original graph and its augmented view generated through graph augmentation. Specifically, given an encoder $Enc(\cdot)$, a readout function $Readout(\cdot)$ and an augmentation function $t(G)$, the training objective of GCL is expressed as:

$$\min_{Enc} \mathcal{L}_{\text{GCL}}(t(G), G, Enc, Readout) \quad (1)$$

$$= -\frac{1}{|\mathcal{G}|} \sum_{n=1}^{|\mathcal{G}|} \left( \log \frac{\exp(\text{sim}(\hat{\mathbf{z}}_n, \mathbf{z}_n))}{\sum_{n' \neq n} \exp(\text{sim}(\hat{\mathbf{z}}_n, \hat{\mathbf{z}}_{n'}))} \right),$$

where $|\mathcal{G}|$ is the number of samples, $\mathbf{z}$ is the graph embedding after encoder and readout functions, $\hat{\mathbf{z}}$ is the embedding of augmented graph, and $\text{sim}(\cdot, \cdot)$ is the similarity function of two embeddings. $\mathcal{L}_{\text{GCL}}$ quantifies the disagreement between graphs.

**Graph Spectrum.** The spectral decomposition of $\mathbf{L}_{\text{norm}}$ is defined as $\mathbf{L}_{\text{norm}} = \text{Lap}(\mathbf{A}) = \mathbf{U}\mathbf{\Lambda}\mathbf{U}^\top$, where the diagonal matrix $\mathbf{\Lambda} = \text{eig}(\text{Lap}(\mathbf{A})) = \text{diag}(\lambda_1, \dots, \lambda_n)$ consists of real eigenvalues $\{\lambda_i | i = 1, \dots, n\}$, known as the graph spectrum. And $\mathbf{U} = [\mathbf{u}_1, \dots, \mathbf{u}_n] \in \mathbb{R}^{n \times n}$ are the corresponding orthonormal eigenvectors known as the spectral bases (Gene et al., 2013).

**Key Protein-Related Terms.** The hierarchical organization of protein structure emerges from the interplay of several fundamental elements. At its core, **amino acid residues** serve as the basic building blocks, with their variable **side chains** determining the chemical properties. These residues are connected by peptide bonds, forming rigid **peptide planes** that include the $C - N$ bond and adjacent atoms ($C_\alpha$, C, N, and O). The spatial arrangement of these planes is governed by **dihedral angles** ($\phi$ and $\psi$), which define the backbone's main conformation and give rise to distinct local conformations along the polypeptide chain. These local conformations are stabilized by **hydrogen bonds** leading to the formation of common **secondary structures** such as $\alpha$-helices and $\beta$-sheets. The combination of these elements determines the overall **protein conformation**, the three-dimensional shape that is essential for protein folding and function, thus influencing classification. The details are presented at Appendix B.

## 4. Methodology

In this section, we introduce two novel graph augmentation methods based on 2D topology-based Functional Community Invariance (FCI) in Sec 4.1 and 3D Protein Structure Invariance (3-PSI) in Sec 4.2. The overview of both augmentation methods is illustrated in Figure 2. Subsequently, we integrate both proposed augmentation methods into a unified GCL framework for protein representation learning in Sec 4.3. The overall architecture is illustrated in Figure 3.

### 4.1. Functional Community Invariance

Within proteins, there are specific regions where residues cluster together to form communities. These clustered residues have similar chemical properties, collaborating to enable specific functions, such as residues clustered around active sites catalyzing enzymatic reactions (Mehta & Beck, 2014). Breaking such clusters during graph augmentation fails to preserve proteins' functionality and thus loses the key information for protein prediction. To address this, we introduce 2D topology-based Functional Community

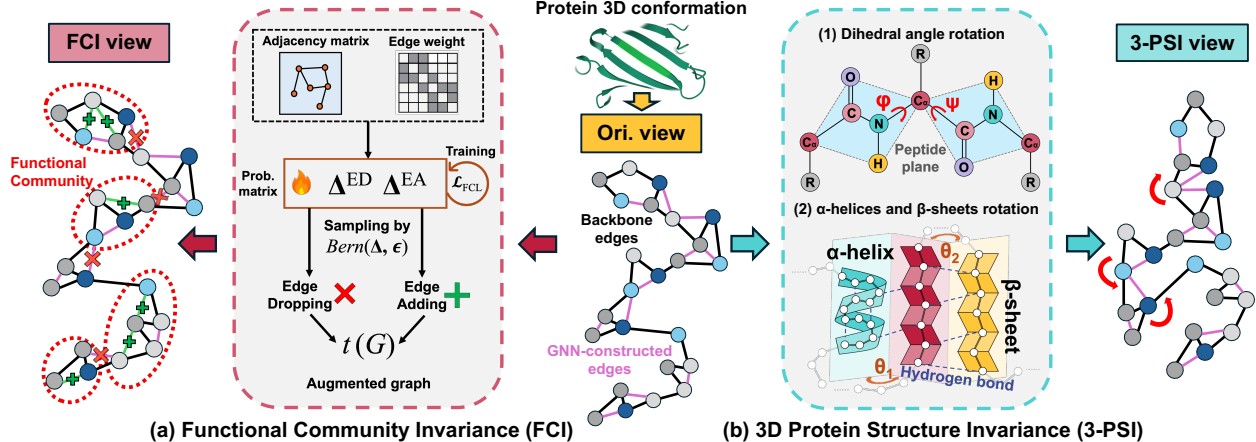

*Figure 2.* This is the overview of our proposed two invariance-based augmentation strategies. (a): FCI takes the adjacency matrix and side-chain similarity as input, optimizing the probability matrix in the spectral space to generate augmentations that preserve functional communities. (b): 3-PSI has two structure-invariance augmentation strategies: (1) Dihedral angle rotation that adjusts $\phi$ and $\psi$ angles to perturb residue coordinates while preserving peptide plane, and (2) $\alpha$-helices and $\beta$-sheets rotation that maintains secondary structures. Middle: The original protein structure with backbone edges and GNN-constructed edges.

Invariance (FCI) to graph augmentation in GCL, which considers both topological structures and chemical properties (Figure 2 (a)), ensuring the preservation of proteins' functional communities during augmentation. In the following paragraphs, we formalize the FCI augmentation as an optimization problem.

**Augmentation Process.** We first formulate the graph augmentation process as sampling from a probability matrix according to a Bernoulli distribution, $t(G) = Bern(\mathbf{\Delta}, \epsilon)$, where $Bern(\cdot) \in \{0, 1\}^{n \times n}$ is the sampling function from Bernoulli distribution, $\mathbf{\Delta} \in [0, 1]^{n \times n}$ is the probability matrix that represents the edge perturbation rates, and $\epsilon$ controls the augmentation strength (the sampled number is less than $\epsilon \cdot m$). By designing a proper probability matrix $\mathbf{\Delta}$, we can ensure that key components of the graph are preserved during the augmentation process.

**CI-based Augmentation.** To ensure the preservation of community structure, we introduce *Community Invariance* (CI)-based augmentation by controlling the spectral changes. According to the upper and lower bounds of spectral changes (Theorem 4.1), maximizing spectral changes involves maximizing their upper bound, which is achieved by flipping edges with largest distance in spectral space. And nodes with larger spectral distances typically belong to different communities (proven in Appendix E.6).

**Theorem 4.1.** *(Bounds of Spectral changes) For a single edge perturbation $A_{ij}$, it induces absolute spectral changes given by $\sum_{y=1}^{n} |\Delta \lambda_y| = \sum_{y=1}^{n} |(U_{iy} - U_{jy})^2 + (\lambda_y - 1)(U_{iy}^2 + U_{jy}^2)|$. It is upper bounded by $\|\mathbf{U}_{i\cdot} - \mathbf{U}_{j\cdot}\|_2^2 + \sum_{y=1}^{n} |\lambda_y - 1|$ and lower bounded by $\|\mathbf{U}_{i\cdot} - \mathbf{U}_{j\cdot}\|_2^2 -$*

$\sum_{y=1}^{n} |\lambda_y - 1|$, *respectively. Here, $\mathbf{U}_{i\cdot}$ represents the $i$-th row vector of $\mathbf{U}$, denoting the $i$-th node embedding in the spectral space.*

Therefore, to preserve communities, we drop edges that cause large spectral changes and add edges that cause small spectral changes, *i.e.*, dropping edges between communities and adding edges within communities. The loss function of Community Invariance is:

$$\max_{\mathbf{\Delta}^{ED}, \mathbf{\Delta}^{EA}} \mathcal{L}_{CI}(\mathbf{\Delta}^{ED}, \mathbf{\Delta}^{EA}) = \mathcal{L}_{ED}(\mathbf{\Delta}^{ED}) - \mathcal{L}_{EA}(\mathbf{\Delta}^{EA}), \quad (2)$$

$$\mathcal{L}_{ED}(\mathbf{\Delta}^{ED}) = \|\mathrm{eig}(\mathrm{Lap}(\mathbf{A} - \mathbf{A} \circ \mathbf{\Delta}^{ED})) - \mathrm{eig}(\mathrm{Lap}(\mathbf{A}))\|_2^2, \quad (3)$$

$$\mathcal{L}_{EA}(\mathbf{\Delta}^{EA}) = \|\mathrm{eig}(\mathrm{Lap}(\mathbf{A} + \mathbf{A}^{\mathbf{c}} \circ \mathbf{\Delta}^{EA})) - \mathrm{eig}(\mathrm{Lap}(\mathbf{A}))\|_2^2, \quad (4)$$

where $\circ$ denotes element-wise product and $\mathbf{A}^c$ is to guide edge addition operations. To achieve CI, we split the probability matrix $\mathbf{\Delta}$ to $\mathbf{\Delta}^{ED}$ and $\mathbf{\Delta}^{EA}$ for edge dropping and edge adding, respectively.

**FCI-based Augmentation.** However, simply preserving communities without considering chemical properties may lead to incorrect assignment of residues with similar functional roles to different communities. To address this issue, we incorporate side-chains into CI-based augmentation, as they reflect the chemical properties of residues (Guy, 1985; Yan & Jernigan, 2005). Specifically, we calculate the similarity between the side-chains of two residues to measure their chemical similarities and use it as the edge weight. We refer to this similarity as *chemical similarity*, which is computed using side-chain embeddings $\mathbf{v}_{sc}$. These embeddings use the sine and cosine values of the first four torsion angles

$\chi_i$ as elements, following Wang et al. (2023b):

$$w_{ij} = \frac{\mathbf{v}_{\text{sc},i} \cdot \mathbf{v}_{\text{sc},j}}{\|\mathbf{v}_{\text{sc},i}\|\|\mathbf{v}_{\text{sc},j}\|}, \quad \mathbf{v}_{\text{sc}} = [\sin(\chi_i), \cos(\chi_i)]_{i=1}^4, \quad (5)$$

where $\chi_{i,1}$ and $\chi_{i,2}$ denote the $i$-th torsion angles of residues $r_i$ and $r_j$, respectively.

**Lemma 4.2.** *For weighted graphs, the absolute spectral changes $\sum_{y=1}^n |\Delta\lambda_y|$ during edge dropping are upper bounded by $|w_{ij}| \cdot \left(\|\mathbf{U}_{i\cdot} - \mathbf{U}_{j\cdot}\|^2 + \sum_{y=1}^n |\lambda_y - 1|\right)$. Here, $w_{ij}$ is the weight of the dropped edge $e_{ij}$.*

By introducing chemical similarity (weights) for edges, clustering communities can better group residues that share similar chemical properties, leading to more biologically meaningful functional communities. According to Lemma 4.2, maximizing the spectral change during edge dropping results in the removal of edges with large edge weights. To ensure that edges with high chemical similarity are preserved while edges crossing functional communities are removed, Function Community Invariance (FCI) is computed as:

$$\max_{\boldsymbol{\Delta}^{\text{ED}}, \boldsymbol{\Delta}^{\text{EA}}} \mathcal{L}_{\text{FCI}}(\boldsymbol{\Delta}^{\text{ED}}, \boldsymbol{\Delta}^{\text{EA}}) = \mathcal{L}'_{\text{ED}}(\boldsymbol{\Delta}^{\text{ED}}) - \mathcal{L}_{\text{EA}}(\boldsymbol{\Delta}^{\text{EA}}), \quad (6)$$

$$\mathcal{L}'_{\text{ED}}(\boldsymbol{\Delta}^{\text{ED}}) = \left\|\boldsymbol{\Delta}^{ED}/\mathbf{W}\right\|^2 \cdot \quad (7)$$
$$\left\|\text{eig}(\text{Lap}(\mathbf{W} - \mathbf{W} \circ \mathbf{A} \circ \boldsymbol{\Delta}^{\text{ED}})) - \text{eig}(\text{Lap}(\mathbf{W}))\right\|_2^2,$$

where the element of $\mathbf{W}$ is chemical similarity defined in Eq. 5, the first term of $\mathcal{L}'_{ED}$ is for preservation of high chemical similarity and the second term is for removal of cross-community edges. $\mathcal{L}_{EA}$ remains the same as in CI since chemical similarity cannot affect edge adding. After solving Eq. 6, the augmented view is sampled by: $t(G) = \mathbf{A} - \mathbf{A} \circ Bern(\boldsymbol{\Delta}^{\text{ED}}, \epsilon) + \mathbf{A}^c \circ Bern(\boldsymbol{\Delta}^{\text{EA}}, \epsilon)$.

### 4.2. 3D Protein Structure Invariance

Exploring diverse 3D protein structures benefits models in understanding protein conformations and enhances the learning of protein representation. However, existing 3D augmentation methods applied for protein graph augmentation, such as ordinate perturbations and anisotropic scaling, disrupt essential structural features, including peptide planes and key secondary structures like $\alpha$-helices and $\beta$-sheets. Consequently, these disruptions in protein structures lead the model to learn inaccurate 3D spatial relationships between residues, degrading its performance on downstream tasks. To alleviate this problem, we propose 3-PSI, a novel graph augmentation method that incorporates two types of 3D protein structure invariance. As illustrated in Figure 2 (b), 3-PSI includes Dihedral Angle Rotation (3-PSIDiag) and $\alpha$-helices and $\beta$-sheets Rotation (3-PSIAlpha). These strategies make use of the flexible and dynamic nature of proteins

in native states (Mishra & Jha, 2022), and reduce disruptions to the primary and secondary structures of proteins, preserving the integrity of proteins as much as possible.

**Dihedral Angle Rotation.** The protein backbone consists of repeating $N - C_\alpha - C$ units, with its folding largely governed by the backbone dihedral angles $\phi$ and $\psi$ (Jha et al., 2005). Directly perturbing absolute atomic coordinates disrupts the integrity of peptide planes and eventually disrupts hydrogen bond networks (Liwo et al., 1993; Nisius & Grzesiek, 2012). To mitigate this issue, we propose 3-PSIDiag, which applies guided and constrained rotations to dihedral angles, preserving the relative integrity of peptide planes. Specifically, we introduce small perturbations $\Delta\phi$ and $\Delta\psi$:

$$\phi^{(\text{new})} = \phi^{(\text{old})} + \Delta\phi, \quad \psi^{(\text{new})} = \psi^{(\text{old})} + \Delta\psi \quad (8)$$

while ensuring that $\left(\phi^{(\text{new})}, \psi^{(\text{new})}\right)$ remains within the allowed Ramachandran region (Ramachandran et al., 1963), thereby maintaining the energetic feasibility of the perturbed protein structure. After perturbing the dihedral angles, the coordinates of residues are adjusted correspondingly. It minimizes disruption to hydrogen bonds and secondary structure interfaces, avoiding generating unrealistic protein structures.

**$\alpha$-helices and $\beta$-sheets Rotation.** Proteins have well-defined secondary structures, primarily $\alpha$-helices and $\beta$-sheets, which maintain protein stability and facilitate essential functions (Stanger et al., 2001). Existing 3D augmentation methods often disrupt the integrity of secondary structures, resulting in the learning of biologically irrelevant protein representations. To address this issue, we introduce 3-PSIAlpha, a controlled rotation method for $\alpha$-helices and $\beta$-sheets to preserve essential secondary structures throughout the augmentation process. Specifically, we first identify the secondary structures in a protein (*i.e.*, $\alpha$-helices and $\beta$-sheets) using the DSSP algorithm (Kabsch & Sander, 1983). Each secondary structure is approximated as a planar region, with its corresponding normal vector denoted as $\mathbf{n}_i$. We define the rotation axis of two adjacent secondary structures as the cross product of their normal vectors:

$$\mathbf{a} = \mathbf{n}_i \times \mathbf{n}_j \quad (9)$$

where $\times$ denotes the cross product operation, ensuring that $\mathbf{a}$ is perpendicular to both planes. A small random rotation angle $\theta$ is then sampled from a uniform distribution to determine the extent of the opening or closing motion around $\mathbf{a}$: $\theta \sim \mathcal{U}(-\theta_{\max}, \theta_{\max})$. This controlled rotation ensures that secondary structures remain intact while introducing realistic conformational variations. A 3D rotation about the axis $\mathbf{a}$ by the angle $\theta$ can be described by the standard Rodrigues rotation matrix $\mathbf{R}_{\mathbf{a},\theta}$. For a unit axis $\widehat{\mathbf{a}} = \frac{\mathbf{a}}{\|\mathbf{a}\|}$, $\mathbf{R}_{\mathbf{a},\theta}$ is given by:

$$\mathbf{R}_{\widehat{\mathbf{a}},\theta} = \mathbf{I}\cos\theta + (1 - \cos\theta)\,\widehat{\mathbf{a}}\,\widehat{\mathbf{a}}^\mathsf{T} + [\widehat{\mathbf{a}}]_\times \sin\theta, \quad (10)$$

where $[\hat{\mathbf{a}}]_\times$ denotes the skew-symmetric matrix, as given in Appendix C.3. Next, we randomly select $n$ secondary structures and rotate them around $\mathbf{a}$ by the angle $\theta$. This ensures that the secondary structure is preserved while still keep meaningful 3D structure perturbations for augmentation.

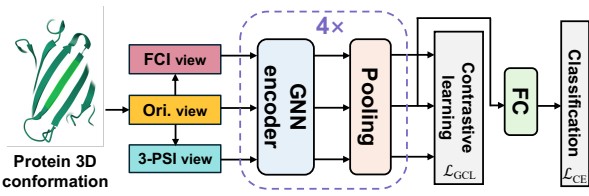

*Figure 3.* The architecture begins with three views of input graphs: the original view and two augmented views (FCI and 3-PSI).

### 4.3. Optimization

We follow the GCL framework introduced in Sec 3, where the original view and augmentation views (FCI and 3-PSI) are passed through the GNN encoder, as illustrated in Figure 3. Following Fan et al. (2023)'s work, each view undergoes four hierarchical pooling operations, with all views sharing the same GNN encoder at each hierarchical level. The output from the final GNN encoder represents each view and is used for contrastive learning. The original view also serves as input for downstream tasks. The GNN encoders are optimized using a combination of GCL loss and classification loss, defined as follows:

$$\mathcal{L}_{\text{ALL}} = \mathcal{L}_{\text{CLS}} + \lambda(\mathcal{L}_{\text{GCL}}^{(\text{FCI})} + \mathcal{L}_{\text{GCL}}^{(\text{3-PSI})}), \qquad (11)$$

where $\mathcal{L}_{\text{CLS}}$ is the classification loss[1], $\mathcal{L}_{\text{GCL}}^{(\cdot)}$ is the GCL loss based on different augmentation methods, and $\lambda$ is the weight to balance the GCL and classification loss. 3-PSI augmentation is selected either by 3-PSIDiag or 3-PSIAlpha. Details of the GNN encoder can be found in Appendix C.4.

## 5. Experiments

### 5.1. Experimental Settings

**Tasks.** Building upon the evaluation protocols of Fan et al. (2023), we assess the effectiveness of our approach across four protein-related tasks: Protein Fold Classification (FOLD), Enzyme Reaction Classification (Reaction), Gene Ontology Term Prediction (GO), and Enzyme Commission (EC) number prediction. For FOLD, we evaluate performance under three scenarios: fold, superfamily, and family classification. For GO, we assess performance across three sub-tasks: biological process (BP), molecular function (MF),

and cellular component (CC) ontology term prediction. Details of tasks and datasets are provided in Appendix D.1.

**Evaluation Metrics.** FOLD and Reaction are single-label classification tasks, where accuracy serves as the evaluation metric. GO and EC are multi-label classification tasks, evaluated using the protein-centric maximum F-score $F_{max}$, as detailed in Appendix D.2.

**Comparison Methods.** We evaluate different combinations of all the proposed augmentation strategies and conduct ablation studies on each strategy to validate their effectiveness. For a comprehensive comparison, we select various SOTA methods from different categories as baselines. **Protein-specific methods**: GVP (Jing et al., 2021), GraphQA (Baldassarre et al., 2020), 3DCNN (Derevyanko et al., 2018), IEConv (Hermosilla & Ropinski, 2022), GearNet (Zhang et al., 2023), ProNet (Wang et al., 2023b), CDConv (Fan et al., 2023); **2D topology-based GCL**: GraphCL (You et al., 2020), Auto-GCL (Yin et al., 2021), MolCLR (Wang et al., 2022b), T-MGCL (Guan & Zhang, 2023) GCL-Span (Lin et al., 2023a), GCS (Wei et al., 2023), CI-GCL (Tan et al., 2024); **3D structure-based GCL**: protein homology modeling tool-based augmentation (Model. Tool) (Waterhouse et al., 2018), and traditional 3D augmentation (Trad. 3D) (Wang et al., 2023b). To our best knowledge, few GCL methods have been specifically designed for proteins. Therefore, we have additionally included two GCL approaches designed for molecules and leveraged their methodological concepts. We adopt a hierarchical Graph Convolutional Network (GCN) serving as the backbone (encoder) for all implemented GCL methods. Details of the baselines and training setup are in Appendix D.3 and C.2, respectively.

### 5.2. Quantitative Results

We summarize the performance comparison of different combinations of FCI and 3-PSI augmentations[2] across four tasks in Table 9. Extra quantitative results are provided in Appendix F. "*Backbone*" is learning without any graph augmentation and only supervised by classification loss.

Compared to 2D topology-based GCL approaches, our proposed FCI consistently enhances performance across all tasks. Notably, it surpasses the best 2D topology-based GCL method, achieving a 4.4% relative gain in GOCC. These results suggest that relying solely on 2D topological information may be insufficient for comprehensive protein representation learning. By preserving functional community information in augmented graphs, FCI enhances protein representation learning, leading to improved performance.

Compared to 3D graph augmentation methods, our proposed 3-PSIDiag and 3-PSIalpha yield consistent improvements

---

[1]Following (Fan et al., 2023), we use NLL loss for single-label protein classification tasks (RC and FOLD) and BCE loss for multi-label protein classification tasks (GO and EC).

[2]Since combining both 3-PSIDiag and 3-PSIAlpha would disrupt the protein structure, we use them separately.

*Table 1.* This table shows the comparison of protein-specific methods, graph contrastive learning approaches (both 2D topology-based and 3D structure-based) and our proposed GCL approach on various protein-related tasks. We report the maximum of F1 score ($F_{max}$) for Enzyme Classification (EC) and Gene Ontology (GO) prediction tasks; and Top-1 accuracy (%) for Fold and Reaction classification tasks. The performance of CDConv is from our reproduction, while other baseline results are from Fan et al. (2023).

| | Method | EC | GO | | | FOLD | | | | Reaction |
| | | | BP | MF | CC | Fold | Super. | Fam. | Avg. | |
|---|---|---|---|---|---|---|---|---|---|---|
| Protein-specific | GVP | 0.489 | 0.326 | 0.426 | 0.420 | 16.0 | 22.5 | 82.8 | 40.4 | 65.5 |
| | 3DCNN | 0.077 | 0.240 | 0.147 | 0.305 | 31.6 | 45.4 | 92.5 | 56.5 | 72.2 |
| | GraphQA | 0.509 | 0.308 | 0.329 | 0.413 | 23.7 | 32.5 | 84.4 | 46.9 | 60.8 |
| | IEConv | - | 0.421 | 0.624 | 0.431 | 47.6 | 70.2 | 99.2 | 72.3 | 87.2 |
| | GearNet-Edge-IEConv | 0.810 | 0.400 | 0.581 | 0.430 | 48.3 | 70.3 | 99.5 | 72.7 | 85.3 |
| | ProNet | - | - | - | - | 52.7 | 70.3 | 99.3 | 74.1 | 86.4 |
| | CDConv | 0.870 | 0.450 | 0.652 | 0.475 | 56.9 | 76.7 | 99.5 | 77.0 | 88.6 |
| 2D Topo. GCL | GraphCL | 0.860 | 0.432 | 0.644 | 0.432 | 56.9 | 77.8 | 99.2 | 77.9 | 86.7 |
| | Auto-GCL | 0.852 | 0.436 | 0.641 | 0.417 | 53.6 | 72.1 | 99.1 | 74.9 | 86.1 |
| | MolCLR | 0.869 | 0.443 | 0.623 | 0.455 | 54.0 | 77.9 | 99.5 | 77.1 | 87.7 |
| | T-MGCL | 0.843 | 0.422 | 0.628 | 0.468 | 55.0 | 77.6 | 99.5 | 77.3 | 85.3 |
| | GCL-Span | 0.859 | 0.434 | 0.650 | 0.419 | 58.2 | 79.1 | 99.4 | 78.9 | 86.1 |
| | GCS | 0.864 | 0.440 | 0.645 | 0.437 | 54.7 | 75.8 | 99.2 | 76.5 | 87.1 |
| | CI-GCL | 0.870 | 0.440 | 0.651 | 0.453 | 57.5 | 79.2 | 99.5 | 78.7 | 86.6 |
| 3D | Trad. 3D Aug. | 0.870 | 0.432 | 0.645 | 0.420 | 56.6 | 78.5 | 99.4 | 78.1 | 86.5 |
| | Auto. Tool Aug. | 0.863 | 0.439 | 0.643 | 0.435 | 56.1 | 77.7 | 99.4 | 77.4 | 86.3 |
| Ours (Ablation) | Backbone | 0.853 | 0.437 | 0.642 | 0.432 | 56.5 | 76.9 | 99.3 | 77.8 | 86.3 |
| | + FCI | 0.878 | 0.446 | 0.656 | 0.473 | 58.7 | 79.4 | 99.5 | 79.2 | 87.2 |
| | + 3-PSIDiag | 0.873 | 0.449 | 0.654 | 0.468 | 58.1 | 78.6 | 99.5 | 78.7 | 86.9 |
| | + 3-PSIAlpha | 0.874 | 0.444 | 0.652 | 0.453 | 56.7 | 80.4 | 99.5 | 78.8 | 88.3 |
| | + 3-PSIDiag + FCI | 0.883 | **0.461** | **0.662** | **0.484** | 58.9 | **81.3** | **99.7** | **80.0** | 87.8 |
| | + 3-PSIAlpha + FCI | **0.885** | 0.454 | 0.659 | 0.477 | **59.8** | 80.8 | 99.5 | **80.0** | **89.0** |

across all tasks. In particular, 3-PSIDiag achieves relative improvements of 11.4% in GOCC while 3-PSIalpha shows relative improvements of 2.1% in Reaction compared with best-performing baselines. These improvements stem from the ability of 3-PSIDiag and 3-PSIAlpha to preserve essential protein structural information in augmented graphs, unlike traditional 3D augmentation methods that risk structural disruption or lack conformational diversity.

Our combined graph augmentation approaches (3-PSIDiag + FCI and 3-PSIAlpha + FCI) achieves substantial improvements over all three categories of baselines. These comprehensive results highlight the synergistic effect of integrating both augmentation strategies, as they complement each other in preserving biological properties of proteins. By maintaining both functional communities and protein 3D structural integrity, our approach generates graphs with enriched protein properties, enabling models to learn better protein representations and enhancing the performance.

## 5.3. Effect of Augmentation Strength

In this section, we evaluate the effect of augmentation strength of our framework in 2D topology-based and 3D structure-based graph augmentations, using the optimal framework setting for each task (*e.g.*, 3-PSIAlpha + FCI for EC, 3-PSIDiag + FCI for GO). The complete results are provided in Appendix F.6.

**2D Augmentation Strength Analysis.** For 2D augmentation comparison, we select GraphCL as the baseline, which randomly drops and adds edges during augmentation. As illustrated in Figure 4, we vary the augmentation strength $\epsilon$ from 0.1 to 0.4, as performance degradation becomes significant beyond this range. Our approach consistently outperforms random augmentation, showing superior performance and stability across different strengths. Furthermore, the results suggest that an optimal $\epsilon$ is critical for generating effective graph views in GCL, as it balances augmentation diversity with the preservation of essential graph properties. The default $\epsilon$ for different tasks is provided in Appendix C.2.

**3D Rotation Strength Analysis.** While our proposed $\alpha$-helices and $\beta$-sheets rotation preserves protein secondary structures, we vary the number of rotation operations from 1 to 5 across different tasks, as the number of rotations directly influences augmentation strength. As illustrated in Figure 5, we observe that FOLD and GOBP achieve

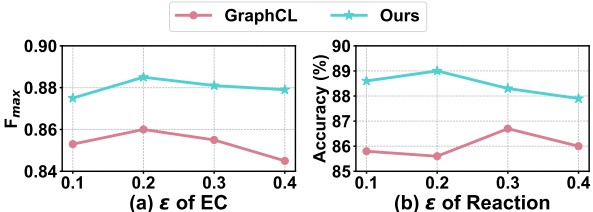

Figure 4. Performance comparison between GraphCL and our approach under different graph augmentation strengths $\epsilon$ (0.1-0.4).

optimal performance with 2 rotations; Reaction and GOCC perform best with 3 rotations. These findings highlight that optimal augmentation strength (2-3 rotations) provides a balance between diversity and fidelity of protein structure. Additionally, the augmentation strength should be selected based on the specific requirements of each task, as too few rotations may not provide sufficient diversity, while rotating excessively can potentially distort the protein structure. The default number of rotations for different tasks is provided in Appendix C.2.

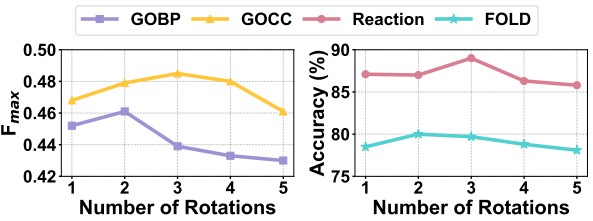

Figure 5. Performance analysis under varying rotational strengths of $\alpha$-helices and $\beta$-sheets across protein-related tasks.

### 5.4. Qualitative Evaluation

In this section, we analyze the robustness of our proposed method, the sensitivity of parameter, and the visualization of representation.

**Robustness Analysis.** In practical application, the protein structures of test data may undergo structural changes in response to environmental factors such as pH, temperature, or the presence of specific ions (Wang et al., 2008). This variability poses a challenge to the model's robustness against structural fluctuations during testing. To systematically evaluate such robustness of our proposed method, we randomly select a proportion (10% to 50%) of residues within each protein sample in the test set and apply rotational transformations to these residues along with their connected segments, mimicking intrinsic structural fluctuations in proteins. Higher proportion of transformed residues represents more significant structural deviations from the initial structure. We compare our method with three strong baselines: Trad. 3D Aug., CDConv and CI-

GCL. As presented in Figure 6, our method consistently outperforms baseline methods across all tasks, particularly at higher proportions, demonstrating strong robustness under high-noise conditions. While all methods experience performance degradation as the proportion increases, our approach exhibits less decline, maintaining $\sim 5\%$ better performance—particularly in the Reaction task—compared to the strongest baseline, Trad. 3D Aug.

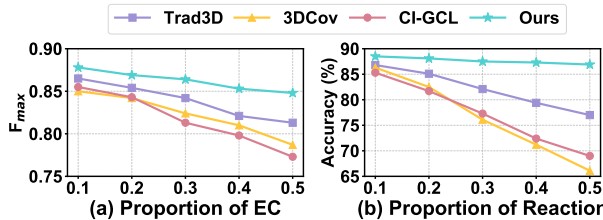

Figure 6. Robustness analysis of models under varying proportion (0.1-0.5) of residues within protein structure. (a) EC, (b) RC.

Recognizing that the resolution of protein data can present a challenge and affect precision, it is crucial to validate the robustness of our approach. Therefore, we provide a specific robustness test designed to evaluate the performance of our method confronted with input protein data of varying resolutions. We follow the resolution classification proposed by Rupp (2009) and divide the test set into low-resolution ($\geq 3$ Å) and high-resolution ($< 3$ Å) structures for three tasks (we cannot retrieve resolution of data in Fold task). Results are displayed in Table 2. The superior performance of our proposed method on low-resolution structures indicates that the model does not overfit to fine-grained coordinate noise but instead learns invariant features critical for function prediction. This behavior demonstrates strong robustness and generalization across inputs of varying quality.

Table 2. Evaluation of robustness to input resolution across various protein function prediction tasks.

| Resolution | EC | GO-BP | GO-MF | GO-CC | Reaction |
|---|---|---|---|---|---|
| Low | **0.903** | **0.485** | 0.648 | **0.504** | **0.921** |
| High | 0.888 | 0.465 | **0.687** | 0.463 | 0.894 |

**Sensitivity Study.** To evaluate the influence of weighting parameter $\lambda$ on model performance, we conducted a sensitivity analysis, the results of which are presented in Table 3. The results demonstrate that model performance exhibits only slight fluctuations across different $\lambda$ values, with $\lambda=1$ generally yielding superior performance on the majority of metrics. This low sensitivity to $\lambda$ indicates that the model is relatively robust to this hyperparameter, suggesting a straightforward optimization process. The complete results are presented in Table 10.

**Visualization of Representation Learning.** We use UMAP

Table 3. Performance of the proposed method using various values for the hyperparameter $\lambda$ in the loss function. Optimal scores are highlighted in bold.

| | $\lambda$ | EC | GO-BP | GO-MF | GO-CC |
|---|---|---|---|---|---|
| Value | 0.2 | 0.882 | 0.451 | 0.656 | 0.467 |
| | 0.6 | 0.879 | 0.443 | 0.658 | 0.479 |
| | 1.0 | **0.885** | **0.461** | **0.662** | 0.484 |
| | 1.4 | 0.880 | 0.457 | 0.649 | **0.489** |
| | 1.8 | 0.877 | 0.448 | 0.653 | 0.475 |

projection (McInnes et al., 2018) to visualize learned representations for the Reaction task, comparing our method with the best-performing baseline (Trad.3D Aug.) to assess representation quality. To demonstrate results clearly, we randomly select 6 classes from Reaction task, each containing more than 20 samples. As visualized in Figure 7, 3-PSIAlpha+FCI exhibits more distinct and compact clusters, with clearer boundaries. In contrast, Trad.3D Aug. shows more scattered distribution patterns with noticeable overlap between classes. Quantitatively, 3-PSIAlpha+FCI achieves both tighter intra-class clustering (distance: 19.90 vs 25.21) and better inter-class separation (distance: 43.33 vs 42.11), indicating its superior protein representation learning.

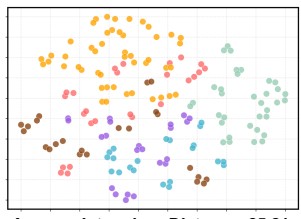 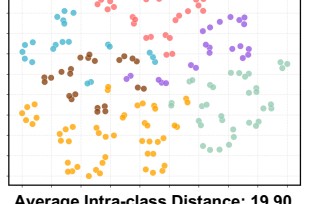

**Average Intra-class Distance: 25.21**
**Average Inter-class Distance: 42.11**
**Trad. 3D Aug.**

**Average Intra-class Distance: 19.90**
**Average Inter-class Distance: 43.33**
**3-PSIAlpha + FCI**

Figure 7. UMAP projection of learned protein representations with quantified intra-class and inter-class distances, comparing 3-PSIAlpha+FCI and Trad.3D Aug. approaches.

### 5.5. Analysis on Complexity

Here we provide the scalability and complexity analysis of our proposed approach. For FCI, preprocessing centers requires computing a probability matrix once, which guides edge perturbations during training. This involves spectral decomposition using the Lanczos algorithm, which efficiently reduces the complexity from $O(N_r^3)$ to $O(N_r K)$, where $N_r$ is the average number of residues per graph and $K$ is the number of selected eigenvalues. With this matrix precomputed, FCI's augmentation during training (which samples edges based on this matrix) can achieve a highly efficient $O(1)$ time complexity. The augmentation operations are accelerated by NumPy's SIMD vectorization. Given that

most of the training time is typically consumed by the GNN encoder, FCI demonstrates good scalability. For 3-PSI, its preprocessing is performed once per protein structure to determine dihedral angles and secondary structures with a time complexity of $O(N_r)$, where $N_r$ is the number of residues. During training, 3-PSI's 3D augmentation perturbs residues, also incurring an $O(N_r)$ time complexity. The memory footprint for this augmentation scales as $O(B \cdot N_r)$, with $B$ denoting the batch size. Despite the linear time complexity for augmentation, these operations benefit significantly from NumPy's SIMD vectorization, leading to fast practical runtime. The resulting performance improvements are considered a justifiable trade-off for additional training time.

### 5.6. Exploration on Ligand Binding Affinity

Since protein structure and functionality are preserved, this motivates an assessment of the model's performance on ligand binding affinity datasets. Here, we briefly explore our best setting model on ligand binding affinity task on PDBbind dataset (Liu et al., 2014). We select a representative baseline Holoprot-Superpixel (Somnath et al., 2021) for comparison and results are represented at Table 4. Results demonstrate that our 3-PSIAlpha + FCI model achieves competitive performance. This superiority underscores the robustness and effectiveness of our approach in learning meaningful protein representations that can be successfully applied to predict complex biomolecular interactions.

Table 4. Model performance comparison on ligand binding affinity task. Model performance comparison. Metrics are abbreviated as follows: R. (RMSE: lower is better), P. (Pearson: higher is better), and S. (Spearman: higher is better). The first three metrics are evaluated at 30% sequence identity, while the last three are at 60%.

| Method | R.↓ | P.↑ | S.↑ | R.↓ | P.↑ | S.↑ |
|---|---|---|---|---|---|---|
| Holoprot-Superpixel | 1.491 | 0.491 | 0.482 | 1.416 | 0.724 | 0.715 |
| 3-PSIAlpha + FCI | **1.462** | **0.515** | **0.510** | **1.383** | **0.753** | **0.749** |

## 6. Conclusion

In this work, we address two key limitations in current GCL approaches for protein representation learning: the lack of biologically meaningful 2D topology-based augmentation and the lack of exploring in 3D protein structural graph augmentation. Specifically, we propose two biology-aware graph augmentation strategies from an invariance perspective: 2D topology-based Functional Community Invariance (FCI) and 3D Protein Structure Invariance (3-PSI), integrating them into a unified GCL framework for protein representation learning. Extensive experiments on four widely used protein benchmarks demonstrate that these biology-aware graph augmentation strategies consistently enhance performance across multiple protein prediction tasks, validating their effectiveness in protein representation learning.

## Impact Statement

This work advances protein representation learning by introducing biologically-aware graph augmentation methods that preserve important structural and functional properties of proteins. We elaborate on the broader impacts of our work from the following two aspects: (1) Better preservation of protein integrity during graph augmentation, reducing the risk of learning biologically implausible representations; (2) A framework that bridges topology-based and structure-based protein analysis, potentially enabling new insights into protein related tasks. This paper presents work whose goal is to advance the field of Machine Learning. There are many potential societal consequences of our work, none which we feel must be specifically highlighted here.

## Acknowledgments

This work was supported in part by Guangdong S&T Program (2021B0909060002) and National Natural Science Foundation of China (NSFC) under Grant No.62204140. We gratefully acknowledge all current related works in this field, which were crucial for completing our research. We also thank Keyu Mao for the discussions on structure retrieval approaches.

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

# A. Augmentation with Homology Modeling

Homology modeling is a computational technique that predicts protein 3D structures by identifying and leveraging structural templates from homologous proteins with known conformations (Chothia & Lesk, 1986). For our implementation, we employ SWISS-MODEL (Waterhouse et al., 2018) for homology modeling and visualize the results using ChimeraX (Meng et al., 2023). The effectiveness of homology modeling varies significantly with template availability. In the upper panel of Figure 8 ('Matched' case), we demonstrate cases where high-identity templates exist (demonstrated with Protein IDs: 1AD6 and 9PCY). Here, we can compare multiple predicted structures against their known original structures, revealing high structural similarity regardless of which template is used. This consistency, while validating the prediction accuracy, also indicates limited structural diversity in the augmented outcomes. The lower panel illustrates a more challenging scenario ('Unmatched' case) using artificially constructed protein sequences that simulate novel proteins without known structures. For each sequence, we designate the prediction from the highest-identity template as a reference structure for comparison. Due to the lack of high-identity templates, these predictions, even when compared to their respective references, exhibit substantial variations in both local secondary structures and global folds, demonstrating the inherent uncertainty in structure prediction when suitable templates are scarce. It also raises concerns about whether such structural uncertainty would impact the reliability and effectiveness of graph augmentation.

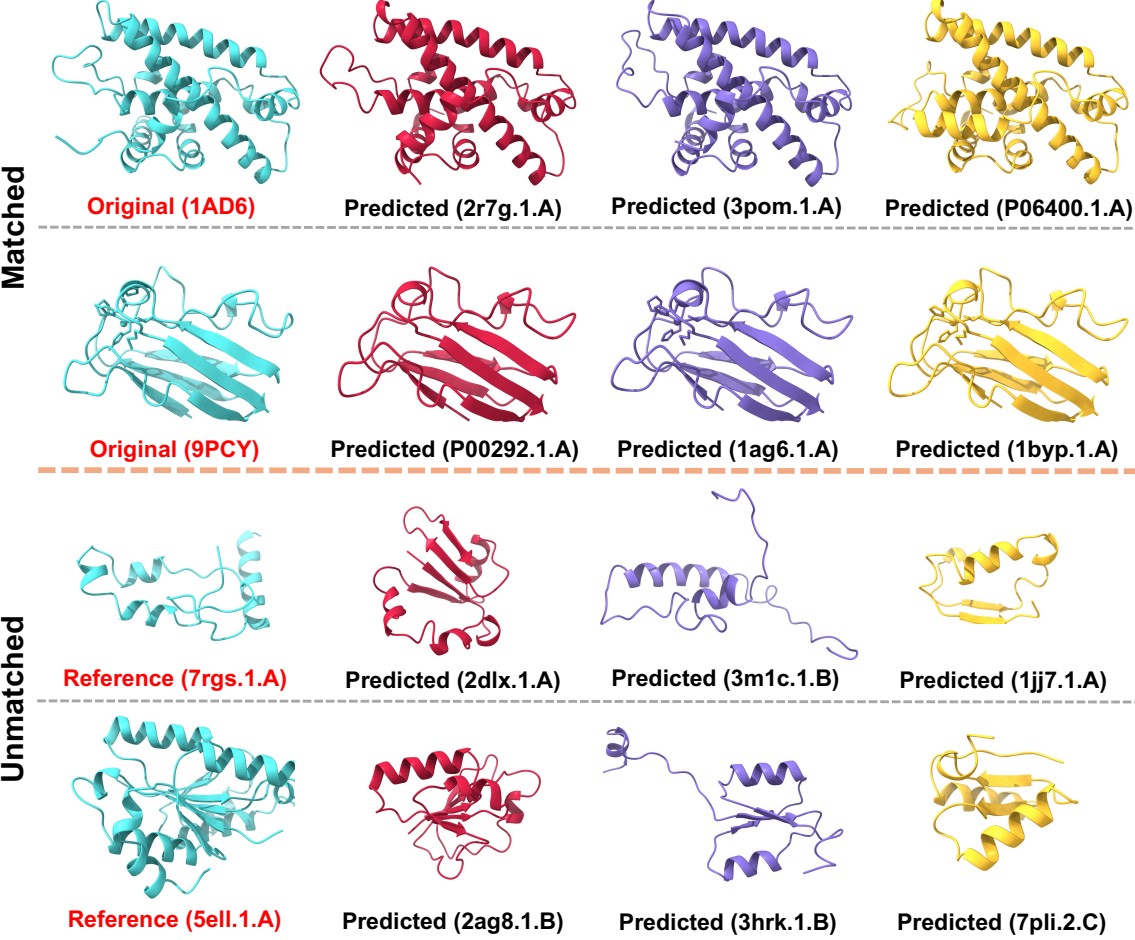

*Figure 8.* Structural variations in SWISS-MODEL predictions. Upper panel: Known proteins (1AD6 and 9PCY) serve as references to evaluate multiple predictions using different templates (template IDs in parentheses), showing high consistency across predictions. Lower panel: For each artificial sequence, the prediction using the highest-identity template is labeled as a reference, with subsequent predictions demonstrating structural diversity arising from template scarcity.

# B. Key Terms Related to Protein

**Amino Acid Residue**: An amino acid residue refers to a single amino acid unit within a polypeptide chain, formed after the loss of water during peptide bond formation. Residues are the basic units of protein sequences and structures, playing a key role in all four tasks: FOLD, Reaction, GO, and EC, as their properties and arrangements determine protein function and classification (Glaser et al., 2001; Doncheva et al., 2011).

**Dihedral Angle**: A dihedral angle (or torsion angle) describes the rotation around a bond in a protein backbone, such as the phi ($\phi$) and psi ($\psi$) angles. These angles define the spatial arrangement of the polypeptide chain. Dihedral angles determine protein folding patterns and are crucial for predicting protein structure in tasks like FOLD and GO (Betancourt & Skolnick, 2004; Wood & Hirst, 2005).

**Hydrogen Bond**: A hydrogen bond is a weak electrostatic interaction between a hydrogen atom (bound to an electronegative atom like nitrogen or oxygen) and another electronegative atom. In proteins, hydrogen bonds stabilize secondary structures like $\alpha$-helices and $\beta$-sheets. Hydrogen bonds are essential for maintaining protein structure and function, directly impacting tasks like FOLD and Reaction (Sticke et al., 1992; Derewenda et al., 1995).

**Peptide Plane**: The peptide plane refers to the rigid, planar structure formed by the peptide bond (C-N) and its adjacent atoms ($C_\alpha$, C, N, and O) in a protein backbone. This planar structure is a fundamental unit of protein conformation. The peptide plane's rigidity influences protein folding and secondary structure formation, which are critical for tasks like FOLD and GO (Roterman, 1995; Hus et al., 2008).

**Protein Conformation**: Protein conformation refers to the three-dimensional shape or spatial arrangement of a protein, determined by the rotation of bonds and interactions between amino acids. Protein conformation is critical for all four tasks (FOLD, Reaction, GO, and EC) as it determines structural, functional, and interaction properties of proteins (Goh et al., 2004; Berkholz et al., 2009).

**Secondary Structure**: The secondary structure of a protein refers to local, repetitive conformations of the polypeptide chain, primarily $\alpha$-helices and $\beta$-sheets, stabilized by hydrogen bonds. Secondary structures are the building blocks of protein folds and are directly relevant to tasks like FOLD and EC, as they influence enzyme active sites and catalytic mechanisms (Jones, 1999; Rost, 2001).

**Side Chain**: The side chain (or R-group) is the variable part of an amino acid that extends from the backbone. Side chains determine the chemical properties and interactions of amino acids. Side chains are critical for protein function, including enzyme catalysis and binding, making them essential for tasks like Reaction and EC (Janin et al., 1978; Maier et al., 2015).

# C. Details of Implementation

## C.1. Graph Construction

The node feature is derived from the Torch Embedding function [3] of residue types. Additionally, we incorporate the side chain embeddings $\mathbf{v}_{sc}$ for each residue to enrich biological information in GNN representations. To achieve **rotational invariance**, we adopt the relative spatial encoding scheme from Fan et al. (2023) to construct edge features. For edge features between nodes $i$ and $j$, we combine sequence information from the original dataset with three spatial components: the normalized relative position vector, its projection in the local coordinate system, and the pairwise distance, ensuring protein structure representation remains invariant under global rotations.

## C.2. Details of Training Setup

Our approach is implemented using the PyTorch framework 2.1.0 and the PyTorch-Geometric library 2.6.0. The CUDA version is 12.4. It is trained on RTX 3090 GPUs. We utilize the SGD optimizer with a momentum of 0.9, a learning rate of 1e-2, and a weight decay of 5e-4. The initial graph construction radius is set to 4. All results are averaged over 5 runs with different random seeds. The training batch sizes for each task are as follows: FOLD (16), Reaction (32), GOCC (32), GOMF (24), GOBP (24), and EC (32). We simply set the weight of GCL loss to lambda=1 in our objective function throughout our experiments. For the augmentation strength of FCI, we set $\epsilon = 0.2$ for GO, EC, and Reaction, and $\epsilon = 0.3$ for FOLD. For the augmentation strength of 3-PSIAlpha, we set 2 rotations for EC, GOBP, and FOLD, and 3 rotations for GOCC and

---

[3]https://pytorch.org/docs/stable/generated/torch.nn.Embedding.html

Reaction. For 3-PSIDiag, we perturb all dihedral angles.

### C.3. Details of Skew-symmetric Matrix

Here is the detail of the skew-symmetric matrix $[\widehat{\mathbf{a}}]_\times$ in Sec 4.2:

$$[\widehat{\mathbf{a}}]_\times = \begin{bmatrix} 0 & -n_z & n_y \\ n_z & 0 & -n_x \\ -n_y & n_x & 0 \end{bmatrix}.$$

### C.4. Details of The Backbone

We implement an Edge-aware Graph Convolutional Network (EdgeGCN) as our GNN encoder. It incorporates edge features into the message-passing framework. The encoder consists of $L$ stacked EdgeGCN layers, where each layer performs the following operations:

$$\mathbf{h}_i^{(l+1)} = \sum_{j \in \mathcal{N}(i)} \mathbf{h}_j^{(l)} \odot \mathbf{W}_e \mathbf{e}_{ij} \tag{12}$$

where $\mathbf{h}_i^{(l)}$ denotes the node features of node $i$ at layer $l$, $\mathcal{N}(i)$ represents the neighborhood of node $i$, $\mathbf{e}_{ij}$ is the edge feature vector between nodes $i$ and $j$, and $\mathbf{W}_e$ is a learnable edge transformation matrix. The operator $\odot$ denotes element-wise multiplication.

We introduce a structure-aware graph pooling mechanism from (Fan et al., 2023) that hierarchically reduces graph resolution while preserving critical structural information in protein graphs. The pooling operation is guided by sequence indices to ensure biologically meaningful node clustering. The pooling process first establishes clustering assignments by halving sequence indices, effectively grouping adjacent nodes in the protein chain. Within each formed cluster, the pooling mechanism aggregates different node attributes through specialized operations: node features and 3D coordinates are averaged to maintain spatial information, orientation vectors are averaged and re-normalized to preserve valid directional information, and sequence information is preserved through maximum pooling. This pooling strategy reduces the graph resolution by approximately half at each level while preserving the essential structural relationships, enabling the network to build hierarchical representations at multiple scales.

## D. Details of Datasets, Evaluation, and Baselines

### D.1. Details of Tasks and Datasets

We use the same dataset settings as the previous work (Fan et al., 2023), and the details are shown as follows:

**Protein Fold Classification.** Protein fold classification is crucial for understanding the relationship between protein structure and evolution. Fold classes capture secondary structure compositions, orientations, and connection orders. Following Hermosilla et al. (2021), we conduct fold classification using the SCOPe 1.75 dataset (Hou et al., 2018), comprising 16,712 proteins across 1,195 fold classes. The 3D coordinates are derived from the SCOPe 1.75 database (Murzin et al., 1995). The dataset provides three evaluation scenarios: **Fold:** Excludes proteins from the same superfamily during training; **Superfamily:** Excludes proteins from the same family during training; **Family:** Includes proteins from the same family during training. Mean accuracy is used as the evaluation metric.

**Enzyme Reaction Classification.** This task involves classifying enzyme-catalyzed reactions based on all four levels of the Enzyme Commission (EC) number (Webb, 1992), representing a protein function classification problem. We utilize the dataset by Hermosilla et al. (2021), which includes 384 four-level Enzyme Commission classes and comprises 29,215/2,562/5,651 proteins for training/validation/test, respectively. Mean accuracy is the evaluation metric.

**Gene Ontology Term Prediction.** This multi-label classification task predicts protein functions through Gene Ontology terms, organized into three hierarchical ontologies: biological process (BP, 1,943 classes), molecular function (MF, 489 classes), and cellular component (CC, 320 classes). Using the dataset from (Gligorijević et al., 2021), we train/validate/test on 29,898/3,322/3,415 proteins, respectively. The $F_{max}$ metric (Fan et al., 2023) is used for evaluation.

**Enzyme Commission Number Prediction.** This multi-label task predicts three-level and four-level EC numbers across 538 classes. Using the dataset from Gligorijević et al. (2021), we train/validate/test on 15,550/1,729/1,919 proteins, respectively. The $F_{max}$ metric is applied for evaluation. For GO term and EC number prediction, we adhere to the multi-cutoff splits from Gligorijević et al. (2021), ensuring the test set only includes PDB chains with a sequence identity $\leq 95\%$ to the training set, consistent with Zhang et al. (2023); Fan et al. (2023).

### D.2. Details of Evaluation Matrix $F_{max}$

Multi-label classification can be interpreted as a set of binary classification tasks. The **protein-centric maximum F-Score** ($F_{max}$), as defined by Gligorijević et al. (2021), is used to evaluate the accuracy of multi-label classification. Given a decision threshold $\lambda \in [0, 1]$, let $p_i^j$ denote the predicted probability for the $j$-th class of the $i$-th protein, $b_i^j \in \{0, 1\}$ be the corresponding binary class label, and $J$ be the total number of classes. The precision and recall for the $i$-th protein are calculated as follows, following the approach described in (Fan et al., 2023):

$$\text{precision}_i(\lambda) = \frac{\sum_{j=1}^{J}(p_i^j \geq \lambda) \cap b_i^j}{\sum_{j=1}^{J}(p_i^j \geq \lambda)}, \quad \text{recall}_i(\lambda) = \frac{\sum_{j=1}^{J}(p_i^j \geq \lambda)}{\sum_{j=1}^{J} b_i^j}.$$

The average precision and recall across all proteins are then defined as:

$$\text{precision}(\lambda) = \frac{\sum_{i=1}^{N} \text{precision}_i(\lambda)}{\sum_{i=1}^{N}\left((\sum_{j=1}^{J}(p_i^j \geq \lambda)) \geq 1\right)}, \quad \text{recall}(\lambda) = \frac{\sum_{i=1}^{N} \text{recall}_i(\lambda)}{N}.$$

Finally, $F_{max}$ is calculated as the maximum F-Score across all thresholds $\lambda \in [0, 1]$:

$$F_{max} = \max_{\lambda \in [0,1]} \left\{ \frac{2 \times \text{precision}(\lambda) \times \text{recall}(\lambda)}{\text{precision}(\lambda) + \text{recall}(\lambda)} \right\}.$$

### D.3. Details of Baselines

For a comprehensive performance evaluation, we compare our approach with the following SOTA baselines, categorized into protein-specific methods, 2D topology-based GCL methods and 3D structure-based GCL:

**Protein-specific methods:**

GVP (Jing et al., 2021) specifically handles scalar and geometric vector features in protein structures, incorporating geometric relationships between amino acids. The method introduces a novel neural network layer that maintains rotational and translational equivariance through specialized vector operations.

GraphQA (Baldassarre et al., 2020) focuses on protein structure quality assessment using graph-based representations of proteins. It uniquely integrates both local and global structural features through a hierarchical graph representation.

3DCNN (Derevyanko et al., 2018) uses 3D convolutional neural networks to analyze protein structures directly in their three-dimensional form. The method employs a voxelization strategy to convert continuous protein structures into discrete 3D grids.

IEConv (Hermosilla & Ropinski, 2022) uses contrastive learning with a specific focus on maintaining geometric invariance and equivariance in protein structure analysis. It introduces specialized convolution operations that preserve geometric symmetries while processing protein structures.

GearNet (Zhang et al., 2023) is specifically designed for protein structure analysis incorporating geometric information. It features a multi-scale message-passing mechanism that captures both short-range and long-range protein interactions.

ProNet (Wang et al., 2023b) is a specialized protein structure learning framework. The framework stands out for its multi-view representation learning approach that jointly considers backbone geometry, side-chain orientations, and residue interactions.

CDConv (Fan et al., 2023) bridges continuous and discrete representations in protein structure analysis. It features an adaptive sampling strategy that efficiently processes both continuous sequences and discrete atom coordinates.

**2D topology-based GCL methods:**

GraphCL (You et al., 2020) is a graph contrastive learning framework that applies various augmentation techniques including node dropping, edge perturbation, attribute masking, and subgraph sampling.

Auto-GCL (Yin et al., 2021) automatically learns optimal augmentation strategies for graph contrastive learning through a bi-level optimization framework.

GCL-Span (Lin et al., 2023a) uses spectral analysis for graph contrastive learning, leveraging graph spectral filtering to create diverse views.

GCS (Wei et al., 2023) focuses on effective sampling strategies with an adaptive mechanism for informative negative samples.

CI-GCL (Tan et al., 2024) incorporates causal inference principles, employing intervention mechanisms to generate counterfactual graph views.

**3D structure-based GCL methods:**,

Traditional 3D augmentation (Wang et al., 2023b), which employs traditional transformations like random coordinate perturbations.

Protein homology modeling tool (Waterhouse et al., 2018) leverages established protein modeling algorithms to generate physically plausible structural variants.

## E. Proofs

### E.1. The Proof of Theorem 4.1 in the Draft

**Definition E.1.** *(Eigenvalue Perturbation)* Assume matrix $\mathbf{A}'$, the altered portion is represented by $\Delta \mathbf{A} = \mathbf{A}' - \mathbf{A}$, and the changed degree is denoted as $\Delta \mathbf{D}$. According to matrix perturbation theory (Hogben, 2013), the change in amplitude for the $y$-th eigenvalue can be represented as:

$$\Delta \lambda_y = \lambda_y' - \lambda_y = \mathbf{u}_y^\top \Delta \mathbf{A} \mathbf{u}_y - \lambda_y \mathbf{u}_y^\top \Delta \mathbf{D} \mathbf{u}_y + \mathcal{O}(\|\Delta \mathbf{A}\|). \tag{13}$$

**Lemma E.2.** *If we only flip one edge $(i, j)$ on adjacency matrix $\mathbf{A}$, the change of $y$-th eigenvalue can be write as*

$$\Delta \lambda_y = \Delta c_{ij} \left( 2 u_{yi} \cdot u_{yj} - \lambda_y \left( u_{yi}^2 + u_{yj}^2 \right) \right), \tag{14}$$

*where $u_{yi}$ is the $i$-th entry of $y$-th eigenvector $\mathbf{u}_y$, and $\Delta c_{ij} = (1 - 2A_{ij})$ indicates the edge flip, i.e $\pm 1$.*

*Proof.* Let $\Delta \mathbf{A}$ be a matrix with only 2 non-zero elements, namely $\Delta A_{ij} = \Delta A_{ji} = 1 - 2A_{ij}$ corresponding to a single edge flip $(i, j)$, and $\Delta \mathbf{D}$ the respective change in the degree matrix, *i.e.* $\mathbf{A}' = \mathbf{A} + \Delta \mathbf{A}$ and $\mathbf{D}' = \mathbf{D} + \Delta \mathbf{D}$.

Denote with $\mathbf{e}_i$ the vector of all zeros and a single one at position $i$. Then, we have $\Delta \mathbf{A} = \Delta c_{ij}(\mathbf{e}_i \mathbf{e}_j^\top + \mathbf{e}_j \mathbf{e}_i^\top)$ and $\Delta \mathbf{D} = \Delta c_{ij}(\mathbf{e}_i \mathbf{e}_i^\top + \mathbf{e}_j \mathbf{e}_j^\top)$.

Based on eigenvalue perturbation formula (13) by removing the high-order term $\mathcal{O}(\|\Delta \mathbf{A}\|)$, we have:

$$\Delta \lambda_y \approx \mathbf{u}_y^\top (\Delta \mathbf{A} - \lambda_y \Delta \mathbf{D}) \mathbf{u}_y \tag{15}$$

Substituting $\Delta \mathbf{A}$ and $\Delta \mathbf{D}$, we conclude Eq. 14. $\square$

**Theorem E.3.** *For unweighted graph, the constraint on the lowest $k$ eigenvalues of the normalized Laplacian matrix $\mathbf{L}_{norm}$ ensures the preservation of the community structure of nodes.*

*Proof.* Firstly, we separate $\Delta \mathbf{A} = \Delta_{\mathbf{A}+} - \Delta_{\mathbf{A}-}$, where $\Delta_{\mathbf{A}+}$ and $\Delta_{\mathbf{A}-}$ indicate which edge is added and deleted, respectively. To analyze the change of eigenvalues in spectral space corresponding to the perturbation of edges in spatial

space, we first consider the situation that only augments one edge for both edge-dropping ($\Delta c_{ij} = -1$) and edge-adding ($\Delta c_{ij} = 1$):

① In the case of edge dropping ($\Delta c_{ij} = -1$ in Lemma E.2), we have

$$\Delta\lambda_y = -2u_{yi} \cdot u_{yj} + \lambda_y \left(u_{yi}^2 + u_{yj}^2\right) \tag{16}$$

$$= (u_{yi} - u_{yj})^2 + (\lambda_y - 1)\left(u_{yi}^2 + u_{yj}^2\right) \tag{17}$$

If we only drop the edge $(i, j)$ that makes a large change in the eigenvalues. We have the objective function as

$$\underset{\{i,j|\Delta c_{ij}=-1\}}{\arg\max} \sum_{y=1}^{n} |\Delta\lambda_y| = |\,(u_{yi} - u_{yj})^2 + (\lambda_y - 1)\left(u_{yi}^2 + u_{yj}^2\right)| \tag{18}$$

$$\leq \sum_{y=1}^{n} |\,(u_{yi} - u_{yj})^2\,| + \sum_{y=1}^{n} |\lambda_y - 1|\left(u_{yi}^2 + u_{yj}^2\right) \tag{19}$$

$$= \|\mathbf{U}_{i\cdot} - \mathbf{U}_{j\cdot}\|^2 + \sum_{y=1}^{n} |\lambda_y - 1|\left(u_{yi}^2 + u_{yj}^2\right) \tag{20}$$

$$\leq \|\mathbf{U}_{i\cdot} - \mathbf{U}_{j\cdot}\|^2 + \sum_{y=1}^{n} |\lambda_y - 1|\left(u_{y1}^2 + u_{y2}^2 + \cdots + u_{yn}^2\right) \tag{21}$$

$$= \|\mathbf{U}_{i\cdot} - \mathbf{U}_{j\cdot}\|^2 + \sum_{y=1}^{n} |\lambda_y - 1| \tag{22}$$

Notice that $\mathbf{u}_y$ is the $y$-th column of $\mathbf{U}$, so $u_{yi} = U_{iy}$ and $\mathbf{U}_{i,\cdot} = [u_{0i}, u_{1i}, \ldots, u_{ni}]$. From the first item in Eq. 35, we prefer to select the nodes with larger distances in the eigenvector spaces, *i.e.* two nodes belonging to different communities (The relationship between eigenvector and community structure is proved in Theorem E.6).

② In the case of edge adding ($\Delta c_{ij} = 1$ in Lemma E.2), we have

$$\Delta\lambda_y = 2u_{yi} \cdot u_{yj} - \lambda_y \left(u_{yi}^2 + u_{yj}^2\right) \tag{23}$$

$$= -(u_{yi} - u_{yj})^2 - (\lambda_y - 1)\left(u_{yi}^2 + u_{yj}^2\right) \tag{24}$$

If we only add the edge $(i, j)$ that makes a small change in the eigenvalues. We have the objective function as

$$\underset{\{i,j|\Delta c_{ij}=+1\}}{\arg\min} \sum_{y=1}^{n} |\Delta\lambda_y| = |\,(u_{yi} - u_{yj})^2 + (\lambda_y - 1)\left(u_{yi}^2 + u_{yj}^2\right)| \tag{25}$$

$$\geq \sum_{y=1}^{n} |\,(u_{yi} - u_{yj})^2\,| - \sum_{y=1}^{n} |1 - (\lambda_y)|\left(u_{yi}^2 + u_{yj}^2\right) \tag{26}$$

$$\geq \|\mathbf{U}_{i\cdot} - \mathbf{U}_{j\cdot}\|^2 - \sum_{y=1}^{n} |(1 - \lambda_y)|\left(u_{y1}^2 + u_{y2}^2 + \cdots + u_{yn}^2\right) \tag{27}$$

$$= \|\mathbf{U}_{i\cdot} - \mathbf{U}_{j\cdot}\|^2 - \sum_{y=1}^{n} |(1 - \lambda_y)| \tag{28}$$

From the first item in Eq. 28, we prefer to select nodes with smaller distances in the eigenvector spaces, *i.e.* nodes belonging to one community (Theorem E.6).

Previously, we have proven that the constraint on the lowest $k$ eigenvalues of $\mathbf{L}$ ensures the preservation of community structure when we only augment one edge. Next, we will demonstrate that the perturbation of more than one edge still aligns with this theory.

Suppose we augment $m$ edges, similar to Lemma 14, we replace $\mathbf{\Delta A} = \sum\limits_{(i,j)\in\{m\text{ edges}\}} \Delta c_{ij}(\mathbf{e}_i\mathbf{e}_j^\top + \mathbf{e}_j\mathbf{e}_i^\top)$, and $\mathbf{\Delta D} = \sum\limits_{(i,j)\in\{m\text{ edges}\}} \Delta c_{ij}(\mathbf{e}_i\mathbf{e}_i^\top + \mathbf{e}_j\mathbf{e}_j^\top)$, we Substituting $\Delta\mathbf{A}$ and $\Delta\mathbf{D}$ of Eq. (31), we get:

$$\Delta\lambda_y = \sum_{(i,j)\in\{m\text{ edges}\}} \Delta c_{ij}\left(2u_{yi}\cdot u_{yj} - \lambda_y\left(u_{yi}^2 + u_{yj}^2\right)\right), \tag{29}$$

By replacing $\Delta c_{ij}\left(2u_{yi}\cdot u_{yj} - \lambda_y\left(u_{yi}^2 + u_{yj}^2\right)\right)$ with Eqs. (35, 28), we could easily see that the community preserving theory is satisfied. $\square$

### E.2. The Proof of Lemma 4.2 in the Draft

**Lemma E.4.** *If we only flip one edge $(i,j)$ on adjacency matrix $\mathbf{A}$ with edge weight $\mathbf{W}$, the change of $y$-th eigenvalue can be write as*

$$\Delta\lambda_y = \begin{cases} -W_{ij}\left(2u_{yi}\cdot u_{yj} - \lambda_y\left(u_{yi}^2 + u_{yj}^2\right)\right), & \text{if edge dropping} \\ 2u_{yi}\cdot u_{yj} - \lambda_y\left(u_{yi}^2 + u_{yj}^2\right), & \text{if edge adding} \end{cases} \tag{30}$$

*where $u_{yi}$ is the $i$-th entry of $y$-th eigenvector $\mathbf{u}_y$.*

*Proof.* Let $\Delta\mathbf{A}$ be a matrix with only 2 non-zero elements, namely $\Delta A_{ij} = \Delta A_{ji} = -W_{ij}$ corresponding to a single edge dropping $(i,j)$, and $\Delta A_{ij} = \Delta A_{ji} = 1$ corresponding to a single edge adding $(i,j)$. $\Delta\mathbf{D}$ is the respective change in the degree matrix, *i.e.* $\mathbf{A}' = \mathbf{A} + \Delta\mathbf{A}$ and $\mathbf{D}' = \mathbf{D} + \Delta\mathbf{D}$.

Denote with $\mathbf{e}_i$ the vector of all zeros and a single one at position $i$. For edge dropping, we have $\Delta_{\mathbf{A}-} = -W_{ij}(\mathbf{e}_i\mathbf{e}_j^\top + \mathbf{e}_j\mathbf{e}_i^\top)$ and $\Delta_{\mathbf{D}-} = -W_{ij}(\mathbf{e}_i\mathbf{e}_i^\top + \mathbf{e}_j\mathbf{e}_j^\top)$. For edge adding, we have $\Delta_{\mathbf{A}+} = (\mathbf{e}_i\mathbf{e}_j^\top + \mathbf{e}_j\mathbf{e}_i^\top)$ and $\Delta_{\mathbf{D}+} = (\mathbf{e}_i\mathbf{e}_i^\top + \mathbf{e}_j\mathbf{e}_j^\top)$.

Based on eigenvalue perturbation formula (13) by removing the high-order term $\mathcal{O}(\|\Delta\mathbf{A}\|)$, we have:

$$\Delta\lambda_y \approx \mathbf{u}_y^\top(\Delta\mathbf{A} - \lambda_y\Delta\mathbf{D})\mathbf{u}_y \tag{31}$$

Substituting $\Delta\mathbf{A}$ and $\Delta\mathbf{D}$, for edge dropping and edge adding respectively, we conclude:

$$\Delta\lambda_y = \begin{cases} -W_{ij}\left(2u_{yi}\cdot u_{yj} - \lambda_y\left(u_{yi}^2 + u_{yj}^2\right)\right), & \text{if edge dropping} \\ 2u_{yi}\cdot u_{yj} - \lambda_y\left(u_{yi}^2 + u_{yj}^2\right), & \text{if edge adding} \end{cases} \tag{32}$$

$\square$

**Lemma E.5.** *For weighted graphs, maximizing the spectral change during edge dropping ensures that edges between communities and those with large edge weights are more likely to be removed.*

*Proof.* For weighted graph (Lemma E.4), we have

$$\Delta\lambda_y = W_{ij}\left(u_{yi} - u_{yj}\right)^2 + W_{ij}(\lambda_y - 1)\left(u_{yi}^2 + u_{yj}^2\right) \tag{33}$$

If we only drop the edge $(i,j)$ that makes a large change in the eigenvalues. We have the objective function (similar to the proof in Theorem E.3) as:

$$\arg\max_{\{i,j\}} \sum_{y=1}^{n} |\Delta\lambda_y| = |W_{ij}| \cdot |\left(u_{yi} - u_{yj}\right)^2 + (\lambda_y - 1)\left(u_{yi}^2 + u_{yj}^2\right)| \tag{34}$$

$$\leq |W_{ij}| \cdot \left(\|\mathbf{U}_{i\cdot} - \mathbf{U}_{j\cdot}\|^2 + \sum_{y=1}^{n}|\lambda_y - 1|\right), \tag{35}$$

In this case, edge $(i,j)$ across different communities ($\|\mathbf{U}_{i\cdot} - \mathbf{U}_{j\cdot}\|^2$ is large) and with large $|W_{ij}|$ are more likely to be removed. $\square$

### E.3. The relation between community and spectrum

**Theorem E.6.** *The spectral decomposition can indicate a relaxed solution of graph vertex partition. Given a graph $G$ with a Laplacian matrix $\mathbf{L}$, and the spectral decomposition is indicated as $\mathbf{D}^{-\frac{1}{2}}\mathbf{L}\mathbf{D}^{-\frac{1}{2}} = \mathbf{U}\mathbf{\Lambda}\mathbf{U}^\top$, where $\mathbf{D}$ is the degree matrix and $\mathbf{U}$, $\mathbf{\Lambda}$ are eigenvectors, eigenvalues correspondingly. The second smallest eigenvalue and its corresponding eigenvector indicate a bipartition of the graph.*

*Proof.* Given a partition of nodes of a graph (split $V$ into two disjoint sets $S_A$ and $S_B$), let $\mathbf{x}$ be an indicator vector for the partition, $x_i = 1$ if node $i$ is in $S_A$, and $-1$, otherwise. Let $d_i$ be the degree of node $i$, and $w_{ij}$ is the weight of edge $(i, j)$. Based on (Shi & Malik, 1997), a normalized cut could be write as

$$Ncut(S_A, S_B) = \frac{\sum_{x_i>0, x_j<0} -w_{ij}x_i x_j}{\sum_{x_i>0} d_i} + \frac{\sum_{x_i<0, x_j>0} -w_{ij}x_i x_j}{\sum_{x_i>0} d_i}. \tag{36}$$

The optimal partition is computed by minimizing the normalized cut: $\min_{\mathbf{x}} Ncut(\mathbf{x})$. By setting $\mathbf{y} = (1 + \mathbf{x}) - b(1 - \mathbf{x})$, $k = \frac{\sum_{x_i>0} d_i}{\sum_i d_i}$ and $b = \frac{k}{1-k}$, we could rewrite it as

$$\min_{\mathbf{x}} Ncut(\mathbf{x}) = \min_{\mathbf{y}} \frac{\mathbf{y}^\top \mathbf{L} \mathbf{y}}{\mathbf{y}^\top \mathbf{D} \mathbf{y}}. \tag{37}$$

with the condition $y_i \in 1, -b$, and $\mathbf{y}^\top \mathbf{D}\mathbf{1} = 0$.

According to the Rayleigh quotient (Gene et al., 2013), we can minimize Eq. 37 by solving the generalized eigenvalue system, if $\mathbf{y}$ is relaxed to take on real values.

$$\mathbf{L}\mathbf{y} = \lambda \mathbf{D}\mathbf{y} \tag{38}$$

By writing $\mathbf{z} = \mathbf{D}^{\frac{1}{2}}\mathbf{y}$, we could transform the Eq. 38 to a standard eigensystem:

$$\mathbf{D}^{-\frac{1}{2}}\mathbf{L}\mathbf{D}^{-\frac{1}{2}}\mathbf{z} = \lambda \mathbf{z} \tag{39}$$

Because the Laplacian matrix is positive semidefinite (Pothen et al., 1990), we can easily verify that $\mathbf{z}_0 = \mathbf{D}^{\frac{1}{2}}\mathbf{1}$ is the smallest eigenvector of Eq. 39 with eigenvalue 0. And correspondingly, $\mathbf{y}_0 = \mathbf{1}$ is the smallest eigenvector with an eigenvalue of 0 in the general eigensystem 38, but do not satisfy the condition $\mathbf{y}^\top \mathbf{D}\mathbf{1} = 0$. According to the Lemma E.7, we could know that the second smallest eigenvector $\mathbf{z}_1$ is the solution of Eq. 37, because $\mathbf{z}_1^\top \mathbf{z}_0 = \mathbf{y}_1^\top \mathbf{D}\mathbf{1} = 0$ satisfy the condition in Eq. 37.

Therefore, the second smallest eigenvalue and its corresponding eigenvector indicate a bipartition of the graph. While the next cut must be perpendicular to others, which is the third smallest eigenvalue corresponding eigenvector $\mathbf{z}_2^\top \mathbf{z}_1 = \mathbf{z}_2^\top \mathbf{z}_0$. Recursively, the $k$-dimensional eigenvectors can represent the community structure of a graph to some extent. □

**Lemma E.7.** *A simple fact about the Rayleigh quotient (Gene et al., 2013): Let $\mathbf{A}$ be a real symmetric matrix. Under the constraint that $\mathbf{x}$ is orthogonal to the $j - 1$ smallest eigenvectors $\mathbf{x}_1, \mathbf{x}_2, ..., \mathbf{x}_{j-1}$, the quotient $\frac{\mathbf{x}^\top \mathbf{A} \mathbf{x}}{\mathbf{x}^\top \mathbf{x}}$ is minimized by the next smallest eigenvector $\mathbf{x}_j$ and its minimum value is the corresponding eigenvalue $\lambda_j$.*

## F. More Detailed Experiments

### F.1. Go and EC Prediction under Different Sequence Cutoffs

For GO and EC, we follow the evaluation protocol established by (Gligorijević et al., 2021), where proteins in the test set are divided based on their sequence similarity to the training set, using cutoffs of 30%, 40%, 50%, 70%, and 95%. This systematic evaluation across different similarity thresholds helps assess model generalization ability and robustness. Table E.3 presents comprehensive results comparing our methods with existing approaches. Our complete framework demonstrates consistent improvements across all four protein-related tasks compared to existing methods. For instance, compared to the top-performing method CDConv, the combination of 3-PSIAlpha and FCI demonstrates superior performance, achieving a 0.0185 improvement at the 95% similarity cutoff in EC. The ablation studies further validate the effectiveness of our design choices, showing that each component contributes to the overall performance improvement. Notably, the performance

*Table 5.* This table shows the comparison of general protein approaches, graph contrastive learning approaches (both 2D topology-based and 3D structure-based) and our proposed GCL approach on Go and EC under different sequence cutoffs. We report the maximum F1 score ($F_{max}$) for Enzyme Classification (EC) and Gene Ontology (GO) prediction tasks; and Top-1 accuracy (%) for Fold and Reaction classification tasks. The performance of CDConv is from our reproduction, while other baseline results are from (Fan et al., 2023).

| | | GO-CC | | | | | GO-MF | | | | |
|---|---|---|---|---|---|---|---|---|---|---|---|
| | Cutoff | 30% | 40% | 50% | 70% | 95% | 30% | 40% | 50% | 70% | 95% |
| Protein-specific | CNN | 0.258 | 0.257 | 0.260 | 0.263 | 0.387 | 0.238 | 0.243 | 0.256 | 0.292 | 0.354 |
| | ResNet | 0.277 | 0.273 | 0.280 | 0.278 | 0.304 | 0.282 | 0.288 | 0.308 | 0.347 | 0.405 |
| | LSTM | 0.263 | 0.264 | 0.269 | 0.270 | 0.283 | 0.223 | 0.229 | 0.245 | 0.276 | 0.321 |
| | Transformer | 0.378 | 0.382 | 0.388 | 0.395 | 0.405 | 0.184 | 0.187 | 0.195 | 0.204 | 0.211 |
| | GearNet | 0.381 | 0.385 | 0.393 | 0.398 | 0.414 | 0.382 | 0.397 | 0.425 | 0.474 | 0.503 |
| | GearNet-Edge | 0.394 | 0.394 | 0.401 | 0.408 | 0.450 | 0.444 | 0.461 | 0.490 | 0.537 | 0.580 |
| | CDConv | 0.429 | 0.433 | 0.442 | 0.447 | 0.475 | 0.535 | 0.551 | 0.573 | 0.620 | 0.652 |
| 2D Topo. GCL | GraphCL | 0.393 | 0.378 | 0.400 | 0.405 | 0.432 | 0.516 | 0.530 | 0.544 | 0.610 | 0.644 |
| | Auto-GCL | 0.373 | 0.374 | 0.399 | 0.394 | 0.417 | 0.510 | 0.531 | 0.556 | 0.604 | 0.641 |
| | GCL-Span | 0.381 | 0.383 | 0.394 | 0.400 | 0.419 | 0.515 | 0.527 | 0.551 | 0.620 | 0.650 |
| | GCS | 0.389 | 0.382 | 0.395 | 0.404 | 0.437 | 0.518 | 0.536 | 0.566 | 0.613 | 0.645 |
| | CI-GCL | 0.357 | 0.373 | 0.391 | 0.410 | 0.453 | 0.519 | 0.533 | 0.561 | 0.614 | 0.651 |
| 3D | Trad. 3D Aug. | 0.385 | 0.384 | 0.387 | 0.394 | 0.420 | 0.521 | 0.543 | 0.547 | 0.615 | 0.645 |
| | Auto. Tool Aug. | 0.367 | 0.376 | 0.380 | 0.405 | 0.435 | 0.515 | 0.540 | 0.535 | 0.612 | 0.643 |
| Ours (Ablation) | Backbone | 0.391 | 0.397 | 0.404 | 0.428 | 0.432 | 0.509 | 0.531 | 0.558 | 0.604 | 0.642 |
| | + FCI | 0.439 | 0.441 | 0.456 | 0.454 | 0.473 | 0.532 | 0.549 | 0.575 | 0.625 | 0.656 |
| | + 3-PSIDiag | 0.438 | 0.437 | 0.445 | 0.456 | 0.468 | 0.532 | 0.552 | 0.578 | 0.620 | 0.654 |
| | + 3-PSIAlpha | 0.425 | 0.429 | 0.431 | 0.437 | 0.453 | 0.525 | 0.544 | 0.572 | 0.618 | 0.652 |
| | + 3-PSIDiag + FCI | **0.459** | **0.462** | 0.467 | **0.468** | **0.484** | **0.542** | **0.559** | **0.585** | **0.629** | **0.662** |
| | + 3-PSIAlpha + FCI | 0.442 | 0.450 | **0.475** | 0.455 | 0.477 | 0.533 | 0.551 | 0.578 | 0.622 | 0.659 |

| | | GO-BP | | | | | EC | | | | |
|---|---|---|---|---|---|---|---|---|---|---|---|
| | Cutoff | 30% | 40% | 50% | 70% | 95% | 30% | 40% | 50% | 70% | 95% |
| Protein-specific | CNN | 0.197 | 0.195 | 0.197 | 0.211 | 0.244 | 0.366 | 0.361 | 0.372 | 0.429 | 0.545 |
| | ResNet | 0.230 | 0.230 | 0.234 | 0.249 | 0.280 | 0.409 | 0.412 | 0.450 | 0.526 | 0.605 |
| | LSTM | 0.194 | 0.192 | 0.195 | 0.205 | 0.225 | 0.247 | 0.249 | 0.270 | 0.333 | 0.425 |
| | Transformer | 0.267 | 0.265 | 0.262 | 0.262 | 0.264 | 0.167 | 0.173 | 0.175 | 0.197 | 0.238 |
| | GearNet | 0.309 | 0.309 | 0.315 | 0.336 | 0.356 | 0.557 | 0.570 | 0.615 | 0.693 | 0.730 |
| | GearNet-Edge | 0.345 | 0.347 | 0.354 | 0.378 | 0.403 | 0.625 | 0.646 | 0.694 | 0.757 | 0.810 |
| | CDConv | 0.380 | 0.387 | 0.402 | 0.431 | 0.450 | 0.721 | 0.748 | 0.782 | 0.825 | 0.870 |
| 2D Topo. GCL | GraphCL | 0.382 | 0.386 | 0.410 | 0.411 | 0.432 | 0.724 | 0.743 | 0.771 | 0.829 | 0.860 |
| | Auto-GCL | 0.365 | 0.373 | 0.385 | 0.408 | 0.436 | 0.714 | 0.746 | 0.783 | 0.834 | 0.852 |
| | GCL-Span | 0.387 | 0.395 | 0.404 | 0.408 | 0.434 | 0.739 | 0.774 | 0.804 | 0.847 | 0.859 |
| | GCS | 0.387 | 0.386 | 0.407 | 0.414 | 0.440 | 0.724 | 0.756 | 0.792 | 0.832 | 0.864 |
| | CI-GCL | 0.392 | 0.391 | 0.411 | 0.420 | 0.440 | 0.749 | 0.783 | 0.810 | 0.844 | 0.870 |
| 3D | Trad. 3D Aug. | 0.378 | 0.389 | 0.388 | 0.405 | 0.432 | 0.747 | 0.760 | 0.796 | 0.843 | 0.870 |
| | Auto. Tool Aug. | 0.368 | 0.375 | 0.383 | 0.397 | 0.439 | 0.729 | 0.761 | 0.788 | 0.839 | 0.863 |
| Ours (Ablation) | Backbone | 0.351 | 0.371 | 0.385 | 0.408 | 0.437 | 0.710 | 0.737 | 0.768 | 0.818 | 0.853 |
| | + FCI | 0.385 | 0.380 | 0.398 | 0.415 | 0.446 | 0.743 | 0.771 | 0.807 | 0.850 | 0.878 |
| | + 3-PSIDiag | 0.377 | 0.385 | 0.394 | 0.412 | 0.449 | 0.744 | 0.769 | 0.806 | 0.852 | 0.873 |
| | + 3-PSIAlpha | 0.381 | 0.380 | 0.390 | 0.413 | 0.444 | 0.738 | 0.765 | 0.803 | 0.848 | 0.874 |
| | + 3-PSIDiag + FCI | **0.391** | **0.399** | **0.410** | **0.436** | **0.461** | 0.745 | **0.783** | **0.815** | 0.858 | 0.883 |
| | + 3-PSIAlpha + FCI | 0.378 | 0.388 | 0.399 | 0.425 | 0.454 | **0.749** | 0.773 | 0.808 | **0.859** | **0.885** |

advantage of our methods is more pronounced at lower sequence similarity cutoffs (30-50%), which represents a more challenging and realistic scenario in protein function prediction, demonstrating remarkable robustness. This suggests that our supervised graph contrastive learning framework better captures fundamental protein structural features, leading to more robust and generalizable predictions.

## F.2. Compare with Protein Language Models

In recent years, protein representation learning has witnessed significant advances through the development of large-scale protein language models and self-supervised learning approaches (Hermosilla & Ropinski, 2022; Zhang et al., 2023). These methods typically rely on extensive pre-training with massive datasets to learn protein representations. For example, ESM-1b (Rives et al., 2019) utilizes UniRef50 (24M sequences), while ProtBERT-BFD (Elnaggar et al., 2020a) leverages the even larger BFD dataset (2.1B sequences). The underlying assumption is that utilizing large-scale protein data during pre-training enables these models to capture fundamental protein properties and patterns. A parallel line of research has explored self-supervised learning specifically for 3D protein structures. These approaches aim to learn effective representations by designing various pretext tasks. For instance, Hermosilla & Ropinski (2022) proposed a contrastive learning framework that maximizes similarity between sub-structures from the same protein while minimizing similarity between sub-structures from different proteins. Zhang et al. (2023) extended this idea by incorporating multiple self-supervised tasks, including: maximizing agreement between different augmented views of the same protein, predicting residue types, and learning to estimate geometric properties.

However, some fundamental questions remain: (1) how can we select self-supervised tasks, which is sufficient to learn truly effective protein representations? (2) these methods typically require extensive computational resources for training on large-scale datasets with data cleaning, preprocessing, and significant effort in designing and validating multiple self-supervised tasks. Different from these works, our work concentrates on designing an effective supervised graph contrastive learning paradigm that enables graph neural networks to better capture protein structural information. Also, this invariant paradigm may improve the baseline for the pre-training or self-supervised learning works. Here, we compare our approach with various SOTA methods including protein language models and self-supervised learning approaches: DeepFRI (Gligorijević et al., 2021), ESM-1b (Rives et al., 2019), ProtBERT-BFD (Elnaggar et al., 2020b), LM-GVP (Wang et al., 2021), residue-level IEConv (Hermosilla & Ropinski, 2022), and GearNet-based methods (Zhang et al., 2023). As demonstrated in Table F.2 and Table F.2, our invariance-based supervised GCL framework achieves comparable or superior performance across multiple tasks, notably outperforming these methods on fold classification and enzyme reaction prediction, despite not using any pre-training or self-supervised learning strategies.

*Table 6.* Accuracy of protein fold classification between different protein language models and our proposed GCL approach. Baseline results are from (Fan et al., 2023).

| Method | Pre-training Dataset (Size) | Fold Classification | | |
|---|---|---|---|---|
| | | Fold | Superfamily | Family |
| DeepFRI | Pfam (10M) | 15.3 | 20.6 | 73.2 |
| ESM-1b | UniRef50 (24M) | 26.8 | 60.1 | 97.8 |
| ProtBERT-BFD | BFD (2.1B) | 26.6 | 55.8 | 97.6 |
| IEConv (residue level) | PDB (476K) | 50.3 | 80.6 | 99.7 |
| GearNet-Edge-IEConv with Multiview Contrast | AlphaFoldDB (805K) | 54.1 | 80.5 | **99.9** |
| GearNet-Edge-IEConv with Residue Type Prediction | AlphaFoldDB (805K) | 48.8 | 71.0 | 99.4 |
| GearNet-Edge-IEConv with Distance Prediction | AlphaFoldDB (805K) | 50.9 | 73.5 | 99.4 |
| GearNet-Edge-IEConv with Angle Prediction | AlphaFoldDB (805K) | 56.5 | 76.3 | 99.6 |
| GearNet-Edge-IEConv with Dihedral Prediction | AlphaFoldDB (805K) | 51.8 | 77.8 | 99.6 |
| **3-PSIDiag + FCI** | - | 58.9 | **81.3** | 99.7 |
| **3-PSIAlpha + FCI** | - | **59.8** | 80.8 | 99.5 |

## F.3. Functional Community Preservation Analysis

In this section, we evaluate how well different 2D topology-based graph augmentation strategies preserve functional communities within protein graphs. Specifically, we focus on protein pockets (*i.e.*, spatially contiguous sets of residues),

*Table 7.* Accuracy of enzyme catalytic reaction classification and $F_{max}$ of gene ontology term prediction and enzyme commission number prediction between different protein language models and our proposed GCL approach. Baseline results are from (Fan et al., 2023).

| Method | Pre-training Dataset (Size) | Enzyme | Gene Ontology | | | Enzyme Commission |
|---|---|---|---|---|---|---|
| | | | BP | MF | CC | |
| DeepFRI* | Pfam (10M) | 63.3 | 0.399 | 0.46 | 0.460 | 0.631 |
| ESM-1b[†] | UniRef50 (24M) | 83.1 | 0.476 | 0.657 | 0.498 | 0.864 |
| ProtBERT-BFD | BFD (2.1B) | 72.2 | 0.279 | 0.456 | 0.408 | 0.838 |
| LM-GVP | UniRef100 (216M) | - | 0.417 | 0.545 | **0.527** | 0.664 |
| IEConv (residue level) | PDB (476K) | 88.1 | 0.468 | 0.661 | 0.516 | - |
| GearNet-Edge with Multiview Contrast | AlphaFoldDB (805K) | 87.5 | **0.490** | 0.654 | 0.488 | **0.874** |
| GearNet-Edge with Residue Type Prediction | AlphaFoldDB (805K) | 86.6 | 0.430 | 0.604 | 0.465 | 0.843 |
| GearNet-Edge with Distance Prediction | AlphaFoldDB (805K) | 87.5 | 0.448 | 0.616 | 0.464 | 0.839 |
| GearNet-Edge with Angle Prediction | AlphaFoldDB (805K) | 86.8 | 0.458 | 0.625 | 0.473 | 0.853 |
| GearNet-Edge with Dihedral Prediction | AlphaFoldDB (805K) | 87.0 | 0.458 | 0.626 | 0.465 | 0.859 |
| **3-PSIDiag + FCI** | - | 87.8 | 0.461 | **0.662** | 0.484 | 0.883 |
| **3-PSIAlpha + FCI** | - | **89.0** | 0.452 | 0.659 | 0.477 | 0.885 |

which naturally align with the radius-based graph construction used in GNNs. Protein pockets directly govern essential protein functions, such as substrate binding, catalysis, and allosteric regulation (Schulze et al., 2016). Although various functional annotation methods exist (such as domain partitioning or coevolutionary residue coupling), these approaches do not necessarily translate into local adjacency in a graph neural network. By contrast, pockets exhibit a spatially contiguous arrangement of residues that naturally aligns with the radius-based graph construction employed by GNNs to some extent. Thus, pockets offer a practical evaluation, unlike other globally defined or purely evolutionary methods that are not readily translated into graph-level representations.

For quantitative evaluation, we first identify functional edges by detecting residues in pockets using fpocket [4]. These edges connecting residues within pockets are considered critical and should be preserved during augmentation. Then we evaluate how well our augmented graphs maintain these essential functional connections by calculating the preservation rate:

Let $\mathcal{E}_{\text{func}}$ be the set of functional edges (connections between residues within identified pockets) in the original graph, and $\mathcal{E}_{\text{aug}}$ be the edges in the augmented graph. The preservation rate $\eta$ is calculated as:

$$\eta = \frac{|\mathcal{E}_{\text{func}} \cap \mathcal{E}_{\text{aug}}|}{|\mathcal{E}_{\text{func}}|} \times 100\% \tag{40}$$

where $|\mathcal{E}_{\text{func}} \cap \mathcal{E}_{\text{aug}}|$ represents the number of functional edges preserved after augmentation, and $|\mathcal{E}_{\text{func}}|$ is the total number of functional edges in the original graph.

As shown in Table 8, our FCI method achieves higher preservation rates across all tasks compared to other methods, demonstrating its effectiveness in maintaining the integrity of functional communities during graph augmentation.

*Table 8.* Preservation rates (%) of functional communities across three graph augmentation methods (Auto-GCL, CI-GCL, and FCI) on four protein-related tasks: GO, EC, FOLD, and Reaction.

| Method | GO | EC | FOLD | Reaction |
|---|---|---|---|---|
| Auto-GCL | 75.29 | 78.93 | 70.88 | 73.43 |
| CI-GCL | 91.87 | 94.21 | 76.88 | 88.05 |
| FCI | 96.69 | 98.02 | 90.87 | 97.45 |

---

[4]https://fpocket.sourceforge.net

## F.4. Compare with Protein Learning Methods

In this section, we further compare with our proposed method with protein structure learning models (PSL): Uni-mol (Zhou et al., 2023), P3G (Huang et al., 2023), ESM-2 (Lin et al., 2023b), S-PLM (Wang et al., 2023a); equivariant graph neural networks (EG) EquiformerV2 (Liao et al., 2024); and structure retrieval approaches (SR) Foldseek (van Kempen et al., 2022). We report results of S-PLM and P3G from their original papers, and ESM-2 from [1]. Uni-mol and EquiformerV2 are reproduced. For the GO task, we evaluated Foldseek on a subset of 150 samples as exploration. Despite this, our method achieves comparable performance, even though Foldseek relies on a larger external protein database (e.g., RCSB and PDB) for retrieval-based prediction. While our GO-BP result is lower, we outperform Foldseek on GO-MF and GO-CC. Trained from scratch on a smaller dataset, our model shows strong generalization and effective representation learning. Uni-mol and EquiformerV2 underperform due to their models being tailored to molecules rather than proteins. Compared to S-PLM, our method achieves competitive results on the EC and GO datasets, while consistently outperforming S-PLM on FOLD and Reaction.

*Table 9.* Evaluation of our proposed method against several categories of protein learning models: protein structure learning (PSL), structure retrieval (SR), and equivariant graph networks (EG). Best scores are highlighted in bold.

| | Method | EC | GO BP | GO MF | GO CC | FOLD Fold | FOLD Super. | FOLD Fam. | FOLD Avg. | Reaction |
|---|---|---|---|---|---|---|---|---|---|---|
| PSL | Uni-mol | 0.721 | 0.347 | 0.441 | 0.397 | 31.4 | 61.0 | 90.5 | 60.9 | 74.2 |
| PSL | P3G | 0.784 | 0.379 | 0.548 | 0.448 | - | - | - | - | - |
| PSL | ESM-2 | 0.861 | 0.460 | 0.663 | 0.427 | 38.5 | **81.5** | 99.2 | 73.0 | - |
| PSL | S-PLM | **0.888** | 0.495 | **0.685** | **0.484** | 37.7 | 78.0 | 98.8 | 71.5 | 86.71 |
| SR | Foldseek | - | **0.582** | 0.570 | 0.472 | - | - | - | - | **90.60** |
| EG | EquiformerV2 | 0.751 | 0.351 | 0.480 | 0.375 | 29.9 | 65.2 | 88.0 | 61.0 | 76.42 |
| | Ours | 0.885 | 0.461 | 0.662 | 0.484 | **59.8** | 81.3 | **99.7** | 80.3 | 89.00 |

## F.5. Sensitive Study of $\lambda$

Complete results of sensitive study are presented at Table 10.

*Table 10.* Performance of the proposed method using various values for the hyperparameter $\lambda$ in the loss function. Optimal scores are highlighted in bold.

| | $\lambda$ | EC | GO BP | GO MF | GO CC | FOLD Fold | FOLD Super. | FOLD Fam. | FOLD Avg. | Reaction |
|---|---|---|---|---|---|---|---|---|---|---|
| Value | 0.2 | 0.882 | 0.451 | 0.656 | 0.467 | 56.5 | 80.2 | **99.8** | 78.8 | 86.2 |
| Value | 0.6 | 0.879 | 0.443 | 0.658 | 0.479 | 58.2 | 80.7 | 99.6 | 79.5 | 87.5 |
| Value | 1.0 | **0.885** | **0.461** | **0.662** | 0.484 | 58.9 | **81.3** | 99.7 | **79.9** | **89.0** |
| Value | 1.4 | 0.880 | 0.457 | 0.649 | **0.489** | **59.1** | 80.6 | 99.7 | 79.8 | 87.8 |
| Value | 1.8 | 0.877 | 0.448 | 0.653 | 0.475 | 58.5 | 80.2 | 99.8 | 79.5 | 88.3 |

## F.6. Complete Results in the Draft

Figure 9 shows results of 2D augmentation strength analysis on all protein-related tasks. For both FOLD and GO, we report the mean performance of their subtasks.

Figure 10 shows results of robustness analysis on GOMF, EC, FOLD and Reaction.

Figure 11 shows results of 3D augmentation strength analysis on all tasks.

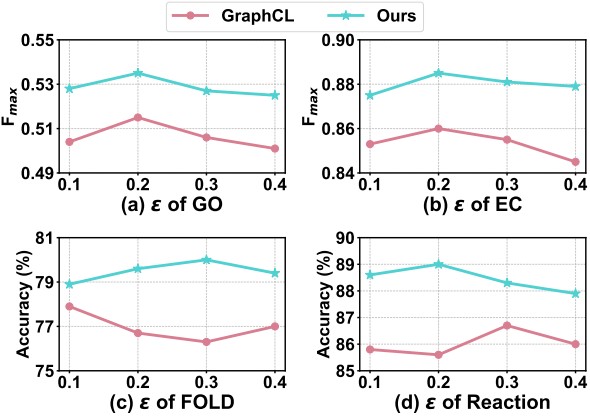

*Figure 9.* Performance comparison between GraphCL and our method under different graph augmentation strengths $\epsilon$ (0.1-0.4). For the GO, we report the average performance of three subtasks.

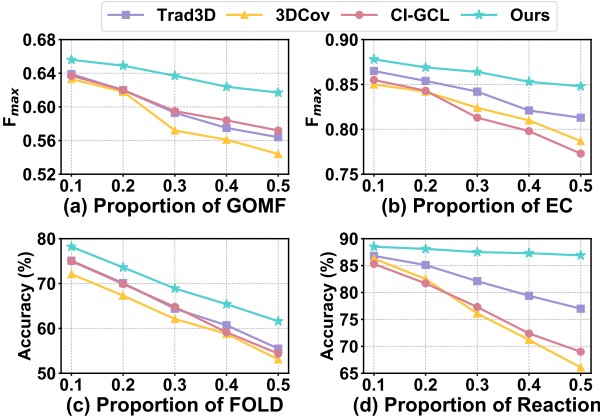

*Figure 10.* Robustness analysis of models under varying perturbation ratios for protein structure. This figure shows model performance ($F_{max}$ for GOMF and EC tasks; accuracy for FOLD and RC tasks) across different noise ratios (0.1-0.5) on four distinct prediction tasks: (a) GOMF, (b) EC, (c) FOLD, and (d) RC.

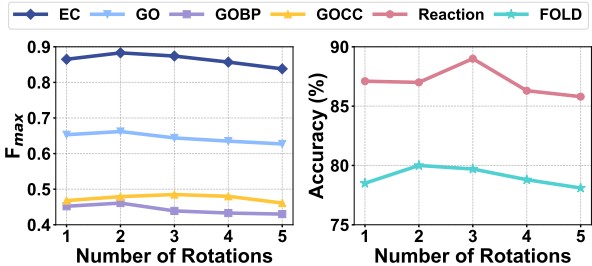

*Figure 11.* Performance analysis of the model under varying $\alpha$-helices and $\beta$-sheets rotational fold strengths across all tasks.

