# OpenReview forum: "Enhancing Graph Contrastive Learning for Protein Graphs from Perspective of Invariance"
_ICML.cc/2025/Conference — ICML 2025 poster_

### Official Review · Reviewer_UCwy · 2025-03-04

**Overall Recommendation:** 3

**Summary:**

This paper proposes a novel framework for protein representation learning by introducing two biologically informed graph-augmentation strategies for contrastive learning. Specifically, it combines:

	1.	Functional Community Invariance (FCI), which preserves crucial residue clusters (communities) involved in protein functionality when augmenting the 2D graph connectivity.

	2.	3D Protein Structure Invariance (3-PSI), which perturbs three-dimensional protein conformations in a biologically plausible manner by manipulating backbone dihedral angles or rotating secondary structures (α-helices and β-sheets) without destroying essential structural motifs.

**Claims And Evidence:**

1. Claim: Biology-Aware Augmentations Improve Representation Quality
	•	The authors propose Functional Community Invariance (FCI) to preserve biologically relevant residue clusters. They measure how often pockets (functional communities) remain intact after augmentation and compare results with standard or random graph augmentations. This directly supports the claim that their method better retains key functional components.
	•	They introduce 3D Protein Structure Invariance (3-PSI) to avoid unrealistic disruptions to three-dimensional protein structures (e.g., by preserving secondary structure integrity or peptide planes).


2. Claim: The Proposed Method Achieves Superior Performance Across Multiple Tasks
	•	The authors conduct evaluations on four tasks: Protein Fold Classification, Enzyme Reaction Classification, Gene Ontology (GO) Prediction, and Enzyme Commission (EC) Number Prediction. Within fold classification and GO, they further break down sub-tasks (e.g., family, superfamily, etc.), generating a broad evidence base.
	•	Results Tables show consistent gains over both:
	•	2D-only GCL baselines (e.g., GraphCL, CI-GCL), and
	•	3D-augmentation baselines (e.g., random coordinate perturbations, homology modeling).
	•	Ablation studies (e.g., using FCI alone, 3-PSI alone, or their combination) confirm that combining the two invariance strategies yields the best improvements.

3. Claim: The Proposed Approach is Robust to Noisy Structural Perturbations
	•	In the robustness experiment, the authors randomly rotate backbone segments in test proteins, simulating environmental or measurement-induced structural variations. They then plot model accuracy/Fmax at proportions from 10% to 50% of residues disturbed.
	•	Results show that their method’s performance degrades more slowly than traditional 3D augmentations and certain baselines, indicating robustness to structural noise.

4. Claim: FCI and 3-PSI are Synergistic
	•	Ablation: FCI or 3-PSI alone improve performance, but together they outperform either approach individually. The results tables consistently show the “FCI + 3-PSI” variant is highest-performing.


The paper’s primary claims are largely well-supported by clear empirical results, multiple tasks, ablation studies, and robust analyses. Minor open questions around comparisons to cutting-edge pretrained models or large-scale data do not undermine the validity of the demonstrated improvements within the scope of graph-based protein learning.

**Essential References Not Discussed:**

1. Rives A, Meier J, Sercu T, Goyal S, Lin Z, Liu J, Guo D, Ott M, Zitnick CL, Ma J, Fergus R. Biological structure and function emerge from scaling unsupervised learning to 250 million protein sequences. Proceedings of the National Academy of Sciences. 2021 Apr 13;118(15):e2016239118.
2. Jumper J, Evans R, Pritzel A, Green T, Figurnov M, Ronneberger O, Tunyasuvunakool K, Bates R, Žídek A, Potapenko A, Bridgland A. Highly accurate protein structure prediction with AlphaFold. nature. 2021 Aug;596(7873):583-9.
3. Batzner S, Musaelian A, Sun L, Geiger M, Mailoa JP, Kornbluth M, Molinari N, Smidt TE, Kozinsky B. E (3)-equivariant graph neural networks for data-efficient and accurate interatomic potentials. Nature communications. 2022 May 4;13(1):2453.

**Experimental Designs Or Analyses:**

1. Benchmark Tasks and Datasets:
The authors evaluate on four widely used tasks in protein representation learning (fold classification, enzyme reaction classification, GO term prediction, and EC number prediction), each with recognized standard datasets.

2. Comparisons with Baselines:
Baseline Variety: The authors compare with multiple 2D topology-based GCL approaches (GraphCL, GCS, etc.) and 3D structure-based augmentations (random coordinate perturbation, homology modeling tools). Hyperparameters: The paper mentions the range of augmentation strengths (e.g., the proportion of edges changed in 2D, or number of dihedral/secondary structure rotations in 3D). They present ablation curves that show how these hyperparameters affect performance. This is helpful to confirm that gains are robust to different augmentation intensities.

3. Ablation Studies:
FCI vs. 3-PSI vs. Combined: The experiments systematically compare using Functional Community Invariance (FCI) alone, 3D Protein Structure Invariance (3-PSI) alone, or combining them. Observing that the combined approach generally yields the best performance supports the synergy claim. Sensitivity to Augmentation Strength: Varying the fraction of edges dropped/added (2D) or the number of secondary-structure rotations/dihedral perturbations (3D) is a direct test of how robust the method is to over- or under-augmentation. This analysis appears well-motivated and thorough.

4. Robustness Checks:
Structural Perturbation at Test Time: The authors apply random rotations to subsets of residues at test time, mimicking protein conformation changes or partial misalignments. They track performance as the proportion of rotated residues increases. This design is a realistic measure of robustness, given that experimental structural data can be noisy.

5. Qualitative Analyses:
Functional Community Preservation: They measure the fraction of intact protein pockets under each augmentation method. This is a direct reflection of how well the augmentation strategy avoids disrupting crucial functional sites.

6. Potential Limitations or Omissions
	1.	Comparison to Protein Language Models: The authors do not evaluate the proposed method in conjunction with or against large-scale pretrained language models (like ESM or ProtT5). While not necessarily invalidating the experiment design, it’s a potential extension.
	2.	Scalability: They do not provide time or memory complexity analyses for 3D augmentations, though that would be helpful for large-scale applications.

**Methods And Evaluation Criteria:**

Method: The paper’s primary contributions—Functional Community Invariance (FCI) and 3D Protein Structure Invariance (3-PSI)—directly address known gaps in protein contrastive learning. Most existing approaches use 2D topology manipulations that may disrupt functionally critical residues or rely on overly simplistic 3D perturbations that distort key structural motifs. By incorporating community preservation (via spectral constraints and side-chain similarity) and 3D backbone constraints (via dihedral angles or secondary structure rotations), the paper’s techniques remain biologically plausible. This stronger biological grounding is exactly what protein graph methods need to capture higher fidelity embeddings.

Benchmark Datasets: The authors run experiments on four major tasks: fold classification, enzyme reaction classification, gene ontology (GO) prediction, and enzyme commission (EC) number prediction. These are widely used benchmarks in structure-based protein modeling research. For fold classification, they even subdivide it (family, superfamily, fold), which is aligned with SCOP classification and demonstrates how well the method distinguishes proteins at varying levels of structural/sequence similarity. GO and EC tasks capture functional annotation challenges, which are biologically relevant real-world scenarios.

Potential Improvements: One area that might strengthen the paper is a scalability discussion—e.g., how the computational overhead of generating more sophisticated 3D augmentations compares with simpler random approaches when moving to large datasets. Another is comparing or combining these graph-based methods with large-scale protein language models. However, the existing baselines and metrics remain representative of graph-based contrastive learning and typical protein structure tasks.

**Other Comments Or Suggestions:**

1.	Highlighting Real-World Applications
	•	The paper demonstrates strong performance on widely used benchmarks, but giving a short paragraph or example on potential practical use cases (e.g., ligand-binding site analysis, enzyme engineering) could clarify why preserving functional communities and realistic 3D geometry might be game-changing in a practical setting.

2.	Discussion on Parameter Sensitivity
	•	There is a helpful analysis of varying augmentation strengths. It might be useful to explicitly mention some rough guidelines for how a user might select these parameters in practice (e.g., typical values of ϵ for edge dropping or typical rotation angles for dihedral modifications).

3.	Further Exploration of Community Detection
	•	The FCI portion uses spectral theory to preserve functional communities. Some readers may appreciate a reference or brief mention of other community detection algorithms (like Louvain or Infomap) or the well-known “normalized cuts” approach to show possible variations or confirm that spectral clustering is a robust choice.

4.	Comparisons to Equivariant Models
	•	Although the authors mention 3D geometry-based GNN approaches, adding one or two lines about E(3)-equivariance or SE(3)-equivariance could clarify that 3-PSI and equivariant GNNs are complementary. They solve different problems (augmentations vs. architecture design) but both preserve geometry in different ways.

**Other Strengths And Weaknesses:**

Additional Strengths

	1.	Originality through Domain-Driven Adaptations
	•	The paper demonstrates a creative fusion of well-known spectral graph theory methods with domain-specific protein structural constraints. While graph contrastive learning itself is not new, the authors’ emphasis on preserving functional communities and realistic 3D transformations elevates the approach beyond standard random augmentations.
	•	By explicitly integrating knowledge about side-chain similarity, functional pockets, dihedral angles, and secondary structure motifs, they bring a strong biological grounding to the contrastive learning framework, which is less common in generic GCL.
	2.	Clarity in Presentation
	•	The paper largely maintains clear structuring: definitions, methodology (with sub-sections for 2D vs. 3D invariance), experiment descriptions, and ablation studies.
	•	Many figures and tables (e.g., pocket-preservation, UMAP visualizations, sensitivity analyses) effectively illustrate key points. This visual clarity helps readers grasp the impact of each augmentation technique.
	3.	Robust Methodological Design
	•	The authors run multiple ablations (e.g., FCI alone, 3-PSI alone, combined) and vary hyperparameters to show where the model sees performance gains. Their robustness experiment specifically showcases how the model handles noisy protein structures—mirroring real-world conditions where data can be imperfect or proteins adopt alternative conformations.

Additional Weaknesses

	1.	Limited Discussion of Scalability
	•	Performing complex 3D transformations (e.g., systematically rotating dihedral angles, secondary structures) might increase computational overhead compared to simpler node/edge manipulations. The paper does not deeply analyze how this scales to massive protein databases, which can be a concern for large-scale or high-throughput applications.
	2.	Comparisons with Non-Graph Methods
	•	The paper positions itself within the ecosystem of graph-based approaches and does a thorough job comparing with various GCL baselines. However, given the explosive growth of large language models for proteins (e.g., ESM), a more explicit recognition or mention of these non-graph methods would help situate the work in the broader landscape of protein representation learning. Without such discussion, some readers might overlook how these approaches could complement or compete with purely sequence-based techniques.
	3.	Dependency on High-Quality 3D Structures
	•	While the approach benefits from realistic structural perturbations, it inherently assumes the existence of reasonably accurate 3D protein models (or experimental structures). In cases where only sequences are available (or structural data is uncertain), the method may not be directly applicable. The paper does not discuss how to handle partial or noisy structural data beyond artificially simulating noise.
	4.	Hyperparameter Tuning Complexity
	•	The method includes several hyperparameters for controlling augmentation strength (fraction of edges dropped, number of secondary structures to rotate, etc.), which might be non-trivial to tune optimally. Although the authors present ablation curves, there is still a risk that real-world users would need extensive trial and error to find the “sweet spot” for a specific application.

**Questions For Authors:**

1. Question: When only partial PDB data is available or some residues are missing coordinates (e.g., unresolved loop regions in cryo-EM structures), how do you apply 3-PSI augmentations? Is there a fallback procedure or do you skip those proteins?

2. Computational Overhead of 3D Augmentations: How does the runtime for dihedral/secondary structure rotations compare with simpler graph augmentations in large-scale experiments? Did you measure any significant slowdown?

3. Criteria for Side-Chain Similarity: In Functional Community Invariance (FCI), side-chain similarity is computed via torsion angles. How robust is this measure for chemically diverse or modified amino acids (e.g., PTMs), and do you incorporate any external chemical knowledge or force-field parameters (e.g., from Amber or CHARMM)?

4. Data Splitting Protocols for Fold Classification: Which SCOP (or SCOPe) version did you use, and what was your sequence identity threshold to ensure minimal overlap between training and test sets?

**Relation To Broader Scientific Literature:**

1.	Protein Representation Learning
	•	Over the past few years, protein representation learning has emerged as a critical area at the intersection of machine learning, structural bioinformatics, and computational biology. Classic approaches include:
	•	Sequence-based methods (e.g., language-model–type architectures such as ESM, ProtT5, and ProtBERT), focusing on large-scale protein sequence corpora. These methods often capture powerful evolutionary context but may overlook critical 3D structural cues.
	•	Structure-based methods (e.g., GNNs like GVP, GearNet, and IEConv), which treat proteins as graphs of residues to incorporate geometric constraints. These approaches have proven effective in tasks such as fold classification, function prediction, and protein–protein interaction studies.

2.	Graph Contrastive Learning (GCL)
	•	In the broader machine learning literature, contrastive learning has become a standard technique for self-supervised representation learning on images, text, and graphs. Recent advances on graphs (e.g., GraphCL, GCS, Auto-GCL, and CI-GCL) demonstrate how data augmentations can push GNNs toward more generalizable embeddings. However, these methods often rely on simplistic graph transformations (like random edge dropping or adding) and rarely leverage specific biological or domain constraints.

3.	Biologically Grounded Augmentations
	•	In protein research, random manipulations of graph structures can cause the loss of essential structural or functional information (e.g., removing edges around catalytic sites). The idea of building biology-aware augmentations partly echoes earlier efforts in structure-based drug design and protein–ligand modeling, where physically plausible perturbations are used to sample conformational states.
	•	This paper’s Functional Community Invariance (FCI) connects with the notion of residue interaction networks and community detection (common in network analyses of protein structures), ensuring that biologically relevant clusters (pockets, communities, etc.) remain intact. Prior literature on “residue co-evolution” and “community detection in protein structures” has highlighted the importance of pockets and functional domains for accurate protein functional interpretation.

**Theoretical Claims:**

The paper’s main theoretical arguments center on characterizing how individual edge perturbations affect the graph spectrum. Specifically:
Theorem 4.1 (Bounds of Spectral Changes)
Claim: When a single edge is flipped (added or removed), the change in the eigenvalues of the normalized Laplacian can be upper-bounded and lower-bounded by expressions involving the spectral embeddings (the eigenvectors) of the unperturbed graph.
Assessment: This result aligns with known techniques in spectral graph theory, where the Davis–Kahan or Weyl inequalities often provide bounds on eigenvalue/eigenvector perturbations. The text’s statement that spectral changes are related to the l_2 distance between the two node eigenvector embeddings also matches typical graph-spectral intuitions: if two nodes lie “far apart” in the spectral embedding, flipping edges between them tends to yield a larger spectral impact. No immediate inconsistencies stand out; the proof outline (in references to standard matrix perturbation results) appears coherent and consistent with existing spectral bounding approaches.

Lemma 4.2 (Weighted Graphs)
Claim: The absolute spectral change when dropping an edge of weight ￼ can be upper-bounded by the product of \(\lvert w_{ij}\rvert\) and terms involving eigenvectors/eigenvalues.
Assessment: This follows a similar logic to Theorem 4.1 but accounts for weighted adjacency. The bound’s proportionality to \(\lvert w_{ij}\rvert\) is intuitively correct—heavier edges should create more significant perturbations when flipped. This is reminiscent of standard matrix perturbation arguments. The statement seems plausible, and no obvious red flags arise in the bounding steps as described. The full formal proofs are outlined briefly in the text (and presumably with additional detail in an appendix). From the information provided and familiarity with spectral graph theory, these claims appear mathematically consistent and do not contradict well-known eigenvalue/eigenvector perturbation theories. The derivations rely on expansions of the difference between Laplacian eigenvalues and standard bounding approaches (e.g., norm-based inequalities). While the paper does not reproduce the entire step-by-step derivation in the main body (likely for brevity), the reasoning is logically sound on inspection.

---

> ### Author Rebuttal · Authors · 2025-03-31
>
> **We sincerely thank reviewer UCwy for detailed reading and meaningful feedback.**
> ***
> > **_Q1_** *Not against large-scale pretrained language models.*
>
> **A1** In Appendix F.2, we have provided comparisons with protein LMs like ESM-1b. For extended experiments, like comparison to ESM-2, you can refer to Reviewer s9oD, A1.
> ***
> > **_Q2_** *Scalability and complexity analysis.*
>
> **A2** Time complexity for 3D augmentation is $O(n)$ for 3-PSI and $O(n^2)$ for FCI, as detailed in A3 in Reviewer s9oD.
> For 3D augmentation, each residue is transformed by a 3×3 rotation matrix. The memory complexity scales as $O(Bn)$, where $B$ is batch size and $n$ is number of residues per sample.
> For large dataset, we can focus augmentation on specific regions of interest (e.g., binding sites) rather than the entire protein to reduce the cost.
> Here we provide running time.
> For 3-PSI, it costs 22.38s and 18.91s each epoch on 3PSI-Diag and 3PSI-Alpha on Fold task.
> Random augmentation (Trad. 3D Aug. in paper) costs 3.56s.
> Overall, Diag and Alpha constitute about 18.0% and 15.3% of total training time (124s per epoch). Considering the performance improvements, the additional training time is a justifiable trade-off.
> ***
> > **_Q3_** *Only sequences are available...& Only partial PDB data is available...*
>
> **A3** We provide possible solutions: for no available experimental structure or PDB data missing, our model can use a predicted structure generated by homology modeling tools such as SWISS-MODEL[1]; for structures that are uncertain with PDB data missing, we can select those from homology modeling tools with higher confidence scores, perform augmentation on each, and then fuse the resulting features.
> As for incomplete PDB data, 3-PSI can also work when a moderate amount of residue remains and you can refer to experiment results in Reviewer s9oD, A2 (incomplete input PDB). If the primary PDB data is unavailable, we need to employ homology modeling tools to reconstruct the structure.
> ***
> > **_Q4_** *Several hyperparameters might be non-trivial to tune optimally & guidelines might needed*
>
> **A4**
> For hyperparameter selection, we follow a common paradigm. Values too small may not provide sufficient augmentation of protein conformation, while larger values may distort it.
> For example, when selecting ε for edge dropping, it is recommended to start with a small value (e.g., 0.1) and gradually adjust it up to 0.4 to avoid damaging the graph structure. For dense graphs, a slightly higher ε can be used, vice versa.
> For ϵ=0.2 and rotation number=2 can serve as a useful starting point for tuning on new datasets.
> ***
> > **_Q5_** *Giving an example on potential practical use...why preserving functional communities...*
>
> **A5** We have provided an experiment of ligand binding affinity task and you can refer to Reviewer ZGC5, A6. Our method achieves competitive performance as it preserves functional communities and protein 3D structure in augmentation, enabling model to learn meaningful representation like structural information.
> ***
> > **_Q6_** *Some readers may appreciate a reference of other community detection algorithms...*
>
> **A6** We will discuss these methods in the main text. Spectral methods can effectively capture edge-level interactions, providing a strong theoretical foundation for maintaining invariance. Therefore, we chose it. We plan to extend FCI to support other related methods.
> ***
> > **_Q7_** *Adding lines about E(3)-equivariance...3-PSI and equivariant GNNs are complementary.*
>
> **A7** 3-PSI focuses on augmenting proteins while preserving key structural properties to generate more biologically reasonable augmentations. In contrast, SE(3) emphasizes the invariance of network inputs to global rotations and translations, ensuring the model's robustness. They are complementary.
> We will discuss them.
> ***
> > **_Q8_** *Is torsion angle-based side-chain similarity robust? Does external chemical used...?*
>
> **A8** Previous work [2] shows that side‐chain torsion angles are effective features of residues. Also, it represents chemical properties of residues. Thus, we compute side‐chain similarity by torsion angles. While our implementation does not incorporate external chemical knowledge or force‐field parameters, we are exploring them to further enhance its robustness, particularly for chemically diverse and modified amino acids.
> ***
> > **_Q9_** *Data Splitting Protocols*
>
> **A9** Following previous work [3], we used SCOP 1.75, filtered with a pairwise identity of less than 95%, and employed a three-level homology reduction dataset as training.
> ***
> **Reference**
>
> [1] Generative models for graph-based protein design
>
> [2] Learning Hierarchical Protein Representations via Complete 3D Graph Networks
>
> [3] DeepSF: deep convolutional neural network for mapping protein sequences to folds
> ***
> **If your concerns have been addressed, could you kindly consider raising your score? We greatly appreciate your comments and support.**

---

### Official Review · Reviewer_ZGC5 · 2025-03-06

**Overall Recommendation:** 3

**Summary:**

This paper improves on the Graph Contrastive Learning (GCL) by introducing two graph augmentation techniques: Functional Community Invariance (FCI) and 3D Protein Structure Invariance (3-PSI). These augmentation techniques are designed to preserve the functional and structural integrity of proteins. The authors used end-to-end training on 4 datasets for classification tasks, integrating contrastive learning loss with classification errors. Experimental results show improvements in classification accuracy and F-1 score, and ablation studies showcase effective preservation of protein structures.

**Claims And Evidence:**

The authors claim that that existing augmentation techniques could lead to incorrect protein structure or disrupt protein functionalities. Such claim is valid and is supported by the design of FCI and 3-PSI augmentation techniques. FCI preserves functional community by controlling spectral changes and incorporating chemical similarity, and 3-PSI preserves the secondary and tertiary structures through controlled rotations of dihedral angles and secondary structures.

**Essential References Not Discussed:**

N/A

**Experimental Designs Or Analyses:**

The authors used 4 datasets for classification tasks only. The model performance and ablation studies on these datasets are valid. However, since the protein structure and functionality is preserved, it is interesting to investigate the model performance on dataset such as ligand binding affinity (eg. PDBbind).

**Methods And Evaluation Criteria:**

The design of the augmentation techniques make sense, as they are rooted in biological principles to ensure that the augmentations are meaningful and realistic. However, the authors did not describe the GNN encoder used in detail, simply pointing to the work of Fan et al. (2023). The model was also trained on an end-to-end fashion for each classification task, which means the learnt representation would change for each downstream task. I have two concerns/suggestions: 1) To fully demonstrate the effectiveness of the proposed augmentation techniques, the authors could use the benchmark models (eg. ProNet, CDConv, GraphCL, GearNet, etc.) and compare performance before and after the augmentation methods are used. 2) The model is trained with equal weight to classification loss and CL loss (λ=1). There is no ablation study on how different λ impact model performance. 3) To demonstrate robustness of the learnt representation, a probable way is to train the GCL model using the proposed augmentation methods and use the learnt representation on downstream tasks.
Regarding evaluation, the proposed metrics (accuracy and F-1 score) is suitable for classification tasks.

**Other Comments Or Suggestions:**

None

**Other Strengths And Weaknesses:**

Strength:
* The manuscript provides a novel approach to integrate domain-specific knowledge in contrastive learning framework.
* The paper is clearly written and well-structured.

Weakness:
* The spectral decomposition might involve high computational cost. The scalability of the proposed approach is not provided.
* The sensitivity of model performance regarding some parameters (eg. λ) is not provided.

**Questions For Authors:**

1. Regarding the GNN encoder structure. In the main text, the authors mentioned that they follow the work of Fan et al. (2023), which is CDConv. However, in Appendix C4, the author mentioned using EdgeGCN with pooling mechanism from Fan’s work. The GNN architecture should be described in the main text, why it is chosen.

2. The augmentation techniques are innovative and biological-aware. However, it would be possible that the augmentation technique works well with the chosen GNN architecture. The authors should eliminate such possibility by applying it on other GNN models, and showcase performance improvements despite the choice of backbone models.

3. Why is the end-to-end training technique chosen for each classification task? Such training loss could lead to representation bias for each task, rather than learning a robust protein representation for multiple downstream tasks.

4. How is the choice of λ in the loss function impact performance?

5. What is the definition of L_(3-PSI)?

**Relation To Broader Scientific Literature:**

The manuscript points out one important limitation in the literature, that the graph augmentation is solely based on topological features, ignoring the intrinsic biological properties of proteins. As one of the most important large molecules to study, how GNN models can be tailored to effectively learn protein representation is an important problem. The proposed augmentation techniques are carefully designed following biological principles and should have broader impact to future work.

**Theoretical Claims:**

Yes, the equations and theorems in the main text are correct. One question, L_FCL is defined but L_(3-PSI) is not clearly defined.

---

> ### Author Rebuttal · Authors · 2025-03-31
>
> **We are grateful to reviewer ZGC5 for insightful reviews.**
> ***
> > **__Q1__** *Did not describe the GNN encoder. In the main text, the authors...in Appendix C4...why GNN is chosen.*
>
> **A1** We sincerely apologize for lack of clarity. We employ GNNs because GNNs can flexibly integrate a protein’s topological and geometric information. Additionally, nodes and edges in the graph can incorporate physicochemical information such as residue distances; then we implement EdgeGCN which fuses edge features into the message-passing framework.
> For Fan et al's work, we employ their hierarchical pooling strategy rather than the encoder. When encoding protein graphs, the hierarchical pooling aggregates nodes layer by layer without losing critical structural information, thereby enabling the GNNs to summarize protein structure at a higher level. Thus, we fuse these components as our encoder. We will provide details in the main text.
> ***
> > **__Q2__** *The model...end-to-end fashion...& Such training lead to bias...& a probable way is to train...*
>
> **A2** We'd like to clarify that our goal is to learn effective protein representations. To this end, we adopt supervised contrastive learning to incorporate label information, which further facilitates representation learning on top of the contrastive framework. Following your suggestion, we conducted additional experiments under a **pure self-supervised** setting. The results show competitive performance, but incorporating supervision further improves results across datasets.
>
> ||EC|GO-BP|GO-MF|GO-CC|FOLD-Fold|FOLD-Super.|FOLD-Family|Reaction|
> |-|-|-|-|-|-|-|-|-|
> |pure self-supervised|0.863|0.443|0.655|0.467|56.9|78.0|99.4|87.2|
> |+ supervised|0.885|0.461|0.662|0.484|59.8|81.3|99.7|89.0|
> ***
> > **__Q3__** *To demonstrate the effectiveness...use the benchmark models and compare.*
>
> **A3** We additionally experimented with two models: CDConv and ProNet with the following results:
>
> ||EC|GO-BP|GO-MF|GO-CC|FOLD-fold|FOLD-Super.|FOLD-Family|Reaction|
> |-|-|-|-|-|-|-|-|-|
> |CDConv|87.9(+0.9)|0.445(-0.005)|0.664(+0.012)|0.479(+0.004)|58.1(+1.2)|81.6(+4.9)|99.8(+0.3)|87.8(-0.8)|
> |Pronet|-|-|-|-|51.8(-0.9)|72.5(+2.3)|99.5(+0.2)|87.1(+0.7)|
>
> where +/- shows performance improvement/reduction.
> After applying our augmentation strategy, performance improved across the majority of tasks, showing its effectiveness.
> ***
> > **__Q4__** *How different λ impact model performance.*
>
> **A4**  We provide an ablation study as follows:
>
> |λ|EC|GO-BP|GO-MF|GO-CC|FOLD-Fold|FOLD-Super.|FOLD-Family|Reaction|
> |-|-|-|-|-|-|-|-|-|
> |0.2|0.882|0.451|0.656|0.467|56.5|80.2|**99.8**|86.2|
> |0.6|0.879|0.443|0.658|0.479|58.2|80.7| 99.6|87.5|
> |1.0|**0.885**|**0.461**|**0.662**|0.484|58.9|**81.3**|99.7|**89.0**|
> |1.4|0.880|0.457|0.649|**0.489**|**59.1**|80.6|99.7|87.8|
> |1.8|0.877|0.448|0.653|0.475|58.5|80.2|99.8|88.3|
>
> The results show that performance fluctuates slightly across different λ values, with λ=1 shows better performances on majority of metrics. It indicates model is not highly sensitive to λ, suggesting an ease of optimization.
> ***
> > **__Q5__** *$L_{\mathrm{FCL}}$ is defined but $L_{\mathrm{3-PSI}}$ is not clearly defined.*
>
> **A5** 3-PSI is used solely for data augmentation based on protein geometry and does not introduce learnable parameters, hence no standalone loss is required. Moreover, the loss defined in Eq.~(11), $L_{\text{GCL}}^{(3\text{-PSI})}$, specifically represents the GCL loss calculated using 3-PSI augmentation.
> ***
> > **__Q6__** *It is interesting to...ligand binding affinity*
>
> **A6** Thanks for your insightful suggestion. We briefly present the results with our best setting on ligand binding affinity task on PDBbind dataset.
> We select a representative baseline [1] for comparison.
>
> ||RMSE↓|Pearson↑|Spearman↑|RMSE↓|Pearson↑|Spearman↑|
> |-|-|-|-|-|-|-|
> |Holoprot-Superpixel|1.491|0.491|0.482|1.416|0.724|0.715|
> |FCI+3PSI-alpha|1.462|0.515|0.510|1.383|0.753|0.749|
>
> Results show that our model achieves competitive performance. The first three metrics are evaluated at **30%** sequence identity, while the last three are at **60%**.
> ***
> > **__Q7__** *The spectral decomposition...high computational cost & The scalability...not provided.*
>
> **A7** The original spectral decomposition has a time complexity of $O(n^3)$ and it is reduced to $O(n^2K)$ by Lanczos algorithm [2], where $n$ is the average number of residues per graph and $K$ is the number of selected eigenvalues. The probability matrix for FCI only needs to be precomputed once for each task. The majority of the training time is attributed to the GNN encoder. Thus, our method is scalable. The running time is detailed in A2 of Reviewer UCwy.
> ***
> **Reference**
>
> [1] Multi-Scale Representation Learning on Proteins
>
> [2] The Lanczos Algorithm with Selective Orthogonalization
> ***
> __If your concerns have been addressed, could you kindly raise the score? We greatly appreciate your comments and support.__

---

### Official Review · Reviewer_s9oD · 2025-03-12

**Overall Recommendation:** 3

**Summary:**

This paper introduces novel biology-aware graph augmentation strategies for protein representation learning within a Graph Contrastive Learning (GCL) framework. The authors identify limitations in existing GCL approaches that either focus exclusively on 2D topology (neglecting intrinsic biological properties) or lack effective 3D structure-based augmentation methods. To address these shortcomings, they develop two complementary strategies: (1) Functional Community Invariance, which preserves topology-driven community structures while incorporating residue-level chemical similarity, and (2) 3D Protein Structure Invariance, which employs dihedral angle perturbations and secondary structure rotations to maintain critical 3D structural information. Experiments across four protein-related tasks demonstrate consistent improvements over existing GCL methods and protein-specific models. The paper offers a valuable solution for GCL in the context of protein structure learning. If the authors could further enhance their main experiments by incorporating comparisons with relevant models, I would be inclined to increase my overall rating accordingly.

**Claims And Evidence:**

The claims are generally well-supported by evidence. The authors provide:

1. Theoretical motivation for both augmentation strategies.

2. Quantitative results showing performance improvements across multiple tasks.

3. Ablation studies demonstrating the contribution of each component.

4. Qualitative analyses showing preservation of functional communities and visualization of learned representations.

The performance improvements are modest but consistent.

**Essential References Not Discussed:**

[r1] Kempen et al. Fast and accurate protein structure search with Foldseek. Nature biotechnology 2024.

[r2] Zhou et al. Uni-mol: A universal 3d molecular representation learning framework. ICLR 2023.

[r3] Huang et al. Protein 3D Graph Structure Learning for Robust Structure-Based Protein Property Prediction. AAAI 2024.

[r4] Lin et al. Evolutionary-scale prediction of atomic-level protein structure with a language model. Science 2023.

[r5] Wang et al. S‐PLM: Structure‐Aware Protein Language Model via Contrastive Learning Between Sequence and Structure. Advanced Science 2025.

[r6] Liao et al. EquiformerV2: Improved Equivariant Transformer for Scaling to Higher-Degree Representations. ICLR 2024.

**Experimental Designs Or Analyses:**

The experimental setup is comprehensive. The authors:

1. Compare against multiple baseline approaches.

2. Perform ablation studies to validate each key component.

3. Analyze augmentation strength effects, evaluate robustness against structural perturbations.

4. Provide qualitative analyses through visualization and functional community preservation.

The performance improvements are modest but consistent across tasks and evaluation settings. The robustness analysis is particularly valuable, showing that their approach maintains better performance under structural perturbations.

**Methods And Evaluation Criteria:**

The proposed methods are appropriate for the problem. The authors evaluate on standard protein-related benchmarks with comprehensive comparison across various baselines, including protein-specific methods, 2D topology-based GCL, and 3D structure-based GCL methods. However:

1. Since protein structures are assumed to be known in advance before model training, their annotations are easily obtainable. Alternatively, structure retrieval approaches like Foldseek [r1] could be used to find similar structures with known function annotations. This should be considered as a baseline in the comparison.

2. The evaluation would benefit from comparison with state-of-the-art structure learning models [r2,r3,r4,r5] and equivariant graph neural networks [r6] as encoders for protein-specific baselines.

3. The paper lacks complexity analysis, making it unclear whether 3-PSIAlpha + FCI could be easily adopted into mainstream transformer-based foundation models.

[r1] Kempen et al. Fast and accurate protein structure search with Foldseek. Nature biotechnology 2024.

[r2] Zhou et al. Uni-mol: A universal 3d molecular representation learning framework. ICLR 2023.

[r3] Huang et al. Protein 3D Graph Structure Learning for Robust Structure-Based Protein Property Prediction. AAAI 2024.

[r4] Lin et al. Evolutionary-scale prediction of atomic-level protein structure with a language model. Science 2023.

[r5] Wang et al. S‐PLM: Structure‐Aware Protein Language Model via Contrastive Learning Between Sequence and Structure. Advanced Science 2025.

[r6] Liao et al. EquiformerV2: Improved Equivariant Transformer for Scaling to Higher-Degree Representations. ICLR 2024.

**Other Comments Or Suggestions:**

Figure 3, Page 5, 'SSI view' -> '3-PSI view'

**Other Strengths And Weaknesses:**

The paper would benefit from deeper comparison with more recent models.

**Questions For Authors:**

1. How does the approach perform when using predicted protein structures (e.g., from AlphaFold) rather than experimental structures? How sensitive is the 3-PSI method to the initial quality of protein structures? Would lower-resolution structures significantly impact performance? This would significantly expand the applicability to proteins without solved structures.

2. Considering the complexity of edge add/drop, can the proposed GCL approaches easily be adopted to more complex encoders?

**Relation To Broader Scientific Literature:**

This paper builds upon both graph contrastive learning and protein-specific approaches. However, the comparison with equivariant graph neural networks and protein foundation models could be more extensive. Recent works like ESM-2 [r1], EquiformerV2 [r2], and other models have shown strong performance on protein-related tasks but are not thoroughly compared against.

[r1] Lin et al. Evolutionary-scale prediction of atomic-level protein structure with a language model. Science 2023.

[r2] Liao et al. EquiformerV2: Improved Equivariant Transformer for Scaling to Higher-Degree Representations. ICLR 2024.

**Theoretical Claims:**

The derivation of the Functional Community Invariance approach is sound, establishing clear connections between spectral constraints and community preservation. The authors effectively develop theoretical links between graph spectra and community structures, providing solid motivation for their approach.

---

> ### Author Rebuttal · Authors · 2025-03-31
>
> **We thank reviewer s9oD for the constructive feedback and valuable suggestions.**
> ***
> > **__Q1__** *Incorporating comparisons with relevant models.*
>
> **A1** We have conducted extensive experiments with the other suggested baselines shown as follows.
>
> |Model|EC|GO-BP|GO-MF|GO-CC|FOLD-fold|FOLD-Super.|FOLD-Family|Reaction|
> |-|-|-|-|-|-|-|-|-|
> |Foldseek|-|**0.582**|0.570|0.472|-|-|-|**90.60**|
> |Uni-mol|0.721|0.347|0.441|0.397|31.35|60.97| 90.51|74.20|
> |S-PLM|**0.888**|0.495|**0.685**|**0.484**|37.74|77.95|98.82| 86.71|
> |P3G [r3]|0.784|0.379|0.548|0.448|-|-|-|-|
> |ESM-2| 0.861|0.460|0.663|0.427|38.50|**81.50**|99.20|-|
> |EquiformerV2|0.751|0.351|0.480|0.375| 29.88|65.18|88.02|76.42|
> |**Ours**|0.885|0.461|0.662 |**0.484**|**59.80** |81.30|**99.70**|89.00|
>
> We report results of S-PLM and P3G from their original papers, and ESM-2 from [1]. Uni-mol and EquiformerV2 are reproduced.
> For the GO task, we evaluated Foldseek on a subset of 150 samples due to time constraints and manual extraction of ground-truth labels. Despite this, our method achieves comparable performance, even though Foldseek relies on a **larger external protein database** (e.g., RCSB and PDB) for retrieval-based prediction. While our GO-BP result is lower, we outperform Foldseek on GO-MF and GO-CC. Trained from scratch on a smaller dataset, our model shows strong generalization and effective representation learning.
> Uni-mol and EquiformerV2 underperform due to their models being tailored to molecules rather than proteins. Compared to S-PLM, our method achieves competitive results on the EC and GO datasets, while consistently outperforming S-PLM on FOLD and Reaction.
> ***
> > **__Q2__** *How does the ... using predicted protein structures? How sensitive is the 3-PSI to the initial quality...? Would lower-resolution structures impact performance?*
>
> **A2** We appreciate your valuable questions. Our approach is designed agnostic to data source and currently being evaluated on experimental structures following previous works.
>
> Regarding the sensitivity of 3-PSI to the initial quality of protein structures, we design two experiments: (1) low resolution (2) partial PDB data available.
>
> (1) Low resolution: We follow the resolution classification proposed by [2] and divide the test set into low-resolution (≥3 Å) and high-resolution (<3 Å) structures for three tasks (we cannot retrieve resolution of data in Fold task). Result shows that 3-PSI maintains strong performance across both resolutions, suggesting that 3-PSI is robust to input resolution.
>
> (2) Partial PDB data available: We simulate incomplete protein structure. Here we remove 20% of residues and train the model. The result is shown in the same table. Here we can find that our model is robust and not sensitive to the initial quality of protein structure.
>
> Overall, our method shows robustness to diverse input qualities, which indicate that our method has the potential of being generalized to predicted protein structures. We will conduct experiments on predicted structures in our future work.
>
> ||EC|GO-BP|GO-MF|GO-CC|Reaction|
> |-|-|-|-|-|-|
> |**Resolution test**|||||||||
> |Low|0.903|0.485|0.648|0.504|0.921|
> |High|0.888|0.465|0.687|0.463|0.894|
> |**Incomplete pdb test**|||||||||
> |Incomplete|0.871|0.453|0.649|0.457|87.6|
> |Complete|0.885|0.461|0.662|0.484|89.0|
> ***
> > **__Q3__** *Complexity analysis and potential of being adopted to more complexed encoders.*
>
> **A3** Our method consists of two stages: preprocessing and training.
> Each protein structure undergoes preprocessing once. For 3-PSI, we iterate over residues to determine the current dihedral angles and secondary structures in $O(n)$ time, where $n$ is the number of residues. For FCI, we search the probability matrix for edge perturbation using the Lanczos algorithm [3] for spectral decomposition, with a time complexity of $O(n^2K)$, where $K$ is the number of selected eigenvalues.
> During training, each 3-PSI augmentation perturbs all residues with $O(n)$ time complexity, while FCI samples edges based on the possibility matrix with $O(n^2)$ complexity; however, both operations benefit significantly from NumPy’s SIMD vectorization, resulting in fast practical runtimes. The running time is detailed in A2 of Reviewer UCwy.
>
> Our method essentially provides an augmentation strategy and is not tightly coupled to specific encoder architecture. It can be extended to more complexed encoders.
> Here we adopt a Graph Transformer encoder and report the performance:
>
> |Model| EC|GO-BP|GO-MF|GO-CC|FOLD-fold|FOLD-Super.|FOLD-Family|Reaction|
> |-|-|-|-|-|-|-|-|-|
> |graph transformer|0.874|0.470|0.652|0.485|58.8|80.0|99.7|87.8|
> ***
> **Reference**
>
> [1] Endowing Protein Language Models with Structural Knowledge
>
> [2] Biomolecular Crystallography: Principles, Practice, and Application to Structural Biology
>
> [3] The Lanczos Algorithm with Selective Orthogonalization
> ***
> **Given these clarifications, could you kindly consider raising your score? We greatly appreciate your support.**

---

> > ### Comment · Reviewer_s9oD · 2025-04-04
> >
> > Thank you for your response. The results of FoldSeek are interesting. As most of my concerns have been addressed, I intend to increase the overall evaluation. The authors should incorporate these valuable discussions and results into the main text of their revised manuscript.

---

> > > ### Author Response · Authors · 2025-04-06
> > >
> > > Thank you for taking the time to review our rebuttal. We sincerely appreciate your insightful feedback and will adjust our manuscript accordingly. Your support is invaluable, and we are confident that our work will make meaningful contributions to the community.

---

### Official Review · Reviewer_exBg · 2025-03-13

**Overall Recommendation:** 3

**Summary:**

This paper investigates methods to improve Graph Contrastive Learning for protein representation learning by incorporating biologically-aware graph augmentation strategies. The authors propose two novel augmentation strategies: Functional Community Invariance (FCI) and 3D Protein Structure Invariance (3-PSI), which are integrated into a unified GCL framework for protein representation learning. Extensive experiments on four protein-related tasks show that their approach consistently improves classification accuracy and robustness over existing 2D topology-based and traditional 3D augmentation methods.

**Claims And Evidence:**

The claims made in the submission are supported by clear and convincing evidence except for the bounds of spectral changes in Theorem 4.1 (line 201 and line 202). The derivation of both the upper and lower bounds appears unclear and requires further clarification.

**Essential References Not Discussed:**

NA

**Experimental Designs Or Analyses:**

I have carefully examined the experimental designs and analyses. Overall, the experiments in this paper are highly convincing. However, when comparing 2D topology-based methods, the selected baseline methods are all graph contrastive learning approaches that do not specifically account for the topological structure of protein graphs. Intuitively, these methods may significantly impact the original protein structure. If there are 2D topology-based methods designed specifically for protein graph topology, it would be beneficial to use them as baselines for comparison.

**Methods And Evaluation Criteria:**

The proposed methods and evaluation criteria make sense for the problem except for the selection of the random rotation angle θ in Section 4.2 (line 246). There appears to be no theoretical justification for the choice of the random rotation angle, which requires further clarification.

**Other Comments Or Suggestions:**

There is an issue in section 4.1 (lemma 4.2, line 209): the weighted upper bound may need to be revised.

**Other Strengths And Weaknesses:**

--Strengths: This paper introduces novel biologically-aware augmentations that explicitly consider both functional and structural integrity as well as improves robustness against structural perturbations, which is crucial for real-world protein modeling. It presents a novel approach to protein modeling, which could significantly inspire protein research and contribute to advancements in drug discovery.

--Weaknesses: Some heuristic choices in augmentation design: The degree of perturbation (ϵ) and number of rotations (θ) remain task-specific hyperparameters, which might require tuning for new datasets.

**Questions For Authors:**

1.	There appears to be no theoretical justification for the choice of the random rotation angle θ in Section 4.2 (line 246), which requires further clarification.
2.	The derivation of both the upper and lower bounds in Theorem 4.1 (line 201 and line 202) appears unclear and requires further clarification.
3.	For 3D-related information of proteins, besides dihedral angles, α-helices, and β-sheets, are there any other factors that may also have a substantial impact on protein structure and function?
4.	Are there 2D topology-based methods designed specifically for protein graph topology? I am convinced that it would be beneficial to use them as baselines for 2D topology-based methods comparison.

**Relation To Broader Scientific Literature:**

1.	This paper extends spectral graph contrastive learning by incorporating functional community constraints, building on prior work like GraphCL, GCS, and CI-GCL.
2.	This paper bridges the gap between 2D and 3D protein graph learning, incorporating structural constraints often overlooked in self-supervised learning.
3.	This paper improves generative augmentation techniques, refining methods used in homology modeling tools (e.g., SWISS-MODEL, MODELLER) by introducing rotation-aware 3D augmentation.

**Theoretical Claims:**

I have verified the accuracy of the claim stated in lines 90–96:
"To generate augmented graphs while preserving the 3D-related information of proteins, 3-PSI employs two distinct coordinate perturbation strategies: (1) rotations of backbone dihedral angles and (2) rotations of secondary structures (α-helices and β-sheets), ensuring that peptide planes and secondary structures remain intact during graph augmentation."
It has been well-established that dihedral angles, α-helices, and β-sheets significantly influence a protein's structure and, consequently, its function. However, it is worth noting that other factors may also have a substantial impact on protein structure and function, which this paper has not considered.

---

> ### Author Rebuttal · Authors · 2025-03-31
>
> **We sincerely appreciate the reviewer exBg's thoughtful and valuable comments.**
> ***
> > **_Q1_** *The derivation of both the upper and lower bounds ... requires clarification. & The weighted upper bound may need to be revised.*
>
> **A1**  We appreciate your valuable feedback on the proof. The proof may be unclear due to the use of different symbols for eigenvalues (k in main text, y in appendix). We will unify them to enhance clarity.
> We clarify the derivation of upper and lower-bounds as follows. When sampling an edge for augmentation, we first determine the change in the y-th eigenvalue of the adjacency matrix by applying the eigenvalue perturbation theory (Lemma E.2, Line 988). Next, we derive the upper and lower bounds by applying the triangle inequality when dropping or adding an edge (see derivations in Theorem E.3, Line 1034 and Line1056, respectively).
>
> Regarding the weighted upper bound, after careful verification, we confirm that no revision is necessary. The change in the eigenvalue for weighted graphs is scaled by the edge weight Wij (Line 1079). By following the same derivation steps as in the unweighted case, we then obtain the upper bound for weighted graphs (Line 1113).
>
> To improve clarity, we will revise the main text to better point readers to the corresponding derivations and proofs in the appendix for the upper and lower bounds.
> ***
> > **_Q2_** *The random rotation angle θ in Section 4.2 (line 246) requires clarification.*
>
> **A2** We appreciate your attention to the selection of the θ. Our goal is to enhance the diversity of protein structures through augmentation while preserving their functionality. Therefore, we prefer to start with relatively small angles. The choice of θ is further guided by our experimental results. We conducted experiments on two tasks with varying θ values. The results show that θ = 10° yields the best performance. Too small θ may lead to insufficient conformational diversity, while a larger θ risks distorting the original protein structure—supporting our hypothesis.
>
> | θ | FOLD-fold | FOLD-Super. | FOLD-Family | Reaction |
> |-------|-----------|-------------|-------------|----------|
> | 5     | 59.5      | 80.5        | 99.7        | 88.8     |
> | **10**| **59.8**  | **81.3**    | **99.7**    | **89.0** |
> | 15    | 59.1      | 80.4        | 99.7        | 88.1     |
> | 20    | 58.6      | 79.9        | 99.4        | 87.4     |
> | 30    | 57.2      | 78.2        | 99.3        | 85.6     |
> | 50    | 56.4      | 77.6        | 98.9        | 83.8     |
> ***
> > **_Q3_** *2D topology-based methods for protein graph topology as baselines for comparison.*
>
> **A3** Thank you for your valuable suggestion. To the best of our knowledge, few GCL methods have been specifically designed for proteins. Therefore, we have additionally included two relevant GCL approaches designed for molecules in the following table.
>
> | Model  |  EC   | GO-BP | GO-MF | GO-CC | FOLD-fold | FOLD-Super. | FOLD-Family | Reaction |
> |--------|-------|-------|-------|-------|-----------|------------|------------|----------|
> |MolCLR[1]|0.869|0.443| 0.623 | 0.455 | 54.0| 77.9       | 99.5       | 87.7     |
> |T-MGCL[2]|0.843|0.422| 0.628 | 0.468 | 55.0      | 77.6       | 99.5       | 85.3     |
> |Ours|**0.885**|**0.461**|**0.662**|**0.484**|**59.8**|**81.3**|**99.7**|**89.0**|
> ***
> > **_Q4_** *Some heuristic choices ... require tuning for new datasets.*
>
> **A4**
> Thank you for your attention. Experiments in Section 5.3 indicate that variations in these hyperparameters do not lead to significant changes in performance, suggesting that our proposed augmentation strategy is relatively robust to their selection. As illustrated in our experiment setting part Appendix C.2., a shared hyper-parameter configuration such as ϵ = 0.2 for GO, EC, and Reaction, and θ = 2 for EC, GOBP, and FOLD, works well across different datasets and tasks. These choices can serve as useful tuning basics for new datasets.
> ***
> > **_Q5_** *Other factors may also have a substantial impact on protein structure and function...*
>
> **A5** We sincerely appreciate the reviewer’s insightful comments.
> We acknowledge that protein structure and function can be influenced by various factors. We primarily use dihedral angles, α-helices and β-sheets rotation to maintain the essential protein structure and improve the protein representation learning. Our experiments have proven the importance of using these factors and keeping invariance in graph augmentation. In future work, we will explore more factors to further strengthen invariance to improve protein representation.
> ***
> **_Reference_**
>
> [1] Molecular Contrastive Learning of Representations via Graph Neural Networks
>
> [2] T-MGCL: Molecule Graph Contrastive Learning Based on Transformer for Molecular Property Prediction
> ***
> __If your concerns have been addressed, could you kindly consider raising your score? We greatly appreciate your comments and support.__

---

### Decision · Program_Chairs · 2025-05-01

**Decision:**

Accept (poster)

**Comment:**

This paper proposes two augmentation strategies for protein graphs: (i) Functional Community Invariance, and (ii) 3D Protein Structure Invariance. The two augmentation strategies are integrated into a Graph Contrastive Learning framework for protein representation learning.

Overall, all reviewers are positive about this work. They generally agree that the proposed augmentations are novel and well motivated. They also agree that the paper is well written and structured. Still, several concerns were raised which revolved around: missing baselines, the scalability of the proposed approach and the lack of complexity analysis, the large number of hyperparameters that need to be tuned, and the dependence of the approach on the availability of 3D structures. Most of these concerns were addressed in the rebuttal, and all four reviewers are leaning towards acceptance. Hence, I recommend the acceptance of this paper. The authors should make sure to update the manuscript to include the reviewers' suggestions.